# OPTIONZERO: PLANNING WITH LEARNED OPTIONS

**Po-Wei Huang[1,2], Pei-Chiun Peng[1,2], Hung Guei[1], Ti-Rong Wu[1][†]**

[1]Institute of Information Science, Academia Sinica, Taiwan
[2]Department of Computer Science, National Yang Ming Chiao Tung University, Taiwan

## ABSTRACT

Planning with options – a sequence of primitive actions – has been shown effective in reinforcement learning within complex environments. Previous studies have focused on planning with predefined options or learned options through expert demonstration data. Inspired by MuZero, which learns superhuman heuristics without any human knowledge, we propose a novel approach, named *OptionZero*. OptionZero incorporates an *option network* into MuZero, providing autonomous discovery of options through self-play games. Furthermore, we modify the dynamics network to provide environment transitions when using options, allowing searching deeper under the same simulation constraints. Empirical experiments conducted in 26 Atari games demonstrate that OptionZero outperforms MuZero, achieving a 131.58% improvement in mean human-normalized score. Our behavior analysis shows that OptionZero not only learns options but also acquires strategic skills tailored to different game characteristics. Our findings show promising directions for discovering and using options in planning. Our code is available at https://rlg.iis.sinica.edu.tw/papers/optionzero.

## 1 INTRODUCTION

Reinforcement learning is a decision-making process in which an agent interacts with environments by selecting actions at each step to maximize long-term rewards. Actions, commonly referred to as *primitive action*, advance the state by one step each. While this granularity allows precise control at each time step, it can lead to inefficiencies in scenarios where predictable sequences of actions are beneficial. For example, in a maze navigation task, it is more efficient to choose a sequence of actions – such as following a straightforward path until reaching a new junction – rather than deciding an action at each time step. This approach reduces the frequency of decision-making and accelerates the learning process. To address these challenges, the concept of *options* (Sutton et al., 1999) has emerged, providing a framework for executing temporally extended actions based on the current state. Options bridge single-step decision-making and strategic long-term planning, not only speeding up the learning process to handle complex scenarios but also simplifying the decision-making by reducing the frequency of choices an agent must consider.

Previous works have proposed adopting the concept of options by either predefining options or learning from expert demonstration data (Sharma et al., 2016; Durugkar et al., 2016; de Waard et al., 2016; Gabor et al., 2019; Czechowski et al., 2021; Kujanpää et al., 2023; 2024). However, the predefined options often rely on a deep understanding of specific environments, and expert data may not be available for every environment, making it difficult to generalize these methods to other environments. Moreover, when planning with options, previous methods require recurrently executing each action within the option to obtain the next states (de Waard et al., 2016) or verifying whether the subgoal can be reached through primitive actions (Czechowski et al., 2021; Kujanpää et al., 2023; 2024). This increases the computational cost when executing longer options during planning, especially in scenarios where environment transitions are expensive.

Inspired by the success of MuZero (Schrittwieser et al., 2020), which employs a learned dynamics network to simulate the environment transitions during planning and achieves superhuman performance from scratch without requiring any human knowledge, this paper proposes a novel approach,

---

[†]Corresponding author: tirongwu@iis.sinica.edu.tw

named *OptionZero*. We modify the MuZero algorithm by integrating an *option network* that predicts the most likely option for each state. During training, OptionZero autonomously discovers options through self-play games and utilizes them during planning, eliminating the requirement for designing options in advance. Furthermore, OptionZero improves the dynamics network to efficiently simulate environment transitions across multiple states with options, significantly reducing the overhead for iterative examination of internal states.

We conduct experiments on Atari games, which are visually complex environments with relatively small frame differences between states, making them suitable for learning options. Our results show that using options with maximum lengths of 3 and 6, OptionZero achieved mean human-normalized scores of 1054.30% and 1025.56%, respectively. In contrast, MuZero achieves a score of only 922.72%. In addition, we provide a comprehensive behavior analysis to examine the options learned and used during planning. Interestingly, the adoption of options varies across different games, aligning with the unique characteristics of each game. This demonstrates that OptionZero effectively discovers options tailored to the specific game states and challenges of each environment. In conclusion, our findings suggest that OptionZero not only discovers options without human knowledge but also maintains efficiency during planning. This makes OptionZero easily applicable to other applications, further extending the versatility of the MuZero algorithm.

## 2 RELATED WORKS

Numerous studies have explored the concepts of *options* in reinforcement learning. For example, de Waard et al. (2016) incorporated options from a predefined option set into Monte Carlo tree search (MCTS) and extended it to focus exploration on higher valued options during planning. Sharma et al. (2016) proposed using two policies for planning: one determines which primitive action to use, and the other determines how many times to repeat that action. Durugkar et al. (2016) explored the effects of repetition and frequency by statistics in Atari games. Vezhnevets et al. (2016) introduced a method which learns options through end-to-end reinforcement learning. Lakshminarayanan et al. (2017) proposed a method that allows agents to dynamically adjust rates of repeated actions. Bacon et al. (2017) derived an option-critic framework, which learns a policy over options and a policy within options. The option policy not only determines how to select and execute an action within options but also learns when to terminate the option. Kim et al. (2023) proposed to adaptively integrate multiple exploration strategies for options based on the option-critic framework. Riemer et al. (2020) introduced a parameter-sharing approach for deep option learning. Young & Sutton (2023) discovered options by learning the option policies and integrated them with a Monte Carlo search. Jinnai et al. (2019) formalized the problem of selecting the optimal option set, and produced an algorithm for discovering the suboptimal option set for planning. Veeriah et al. (2021) proposed a meta-gradient approach for discovering reusable, task-independent options. In addition, several works have studied subgoals, which represent a target state to achieve after several time steps, either segmented by predefined time step intervals or predicted dynamically by a learned network. For example, Gabor et al. (2019) used predefined subgoals for planning in MCTS. Czechowski et al. (2021) introduced a Subgoal Search method to obtain fixed-length subgoals with a low-level policy that predicts primitive actions for reaching subgoals. Kujanpää et al. (2023) proposed Hierarchical Imitation Planning with Search (HIPS), which learns subgoals from expert demonstration data. Kujanpää et al. (2024) extended HIPS to HIPS-$\epsilon$, adding a low-level (primitive action) search to the high-level (subgoal) search, guaranteeing that subgoals are reachable. In summary, these previous works either adopt predefined options, learn subgoals from expert data, or do not incorporate options in MCTS planning. Compared to these works, our goal is to automatically discover options without relying on predefined options or expert data and to use options during planning.

## 3 MUZERO

MuZero (Schrittwieser et al., 2020) is based on the foundation of AlphaZero (Silver et al., 2018), distinguishing itself by learning environment transitions using neural networks. This allows MuZero to plan in advance without extra interaction with the environment, which is particularly advantageous in environments where such interactions are computationally expensive. Consequently, MuZero has achieved success in a wide range of domains (Schrittwieser et al., 2020; Danihelka et al., 2022; Antonoglou et al., 2021; Hubert et al., 2021; Mandhane et al., 2022; Wang et al., 2023).

For planning, MuZero adopts Monte Carlo tree search (MCTS) (Kocsis & Szepesvári, 2006; Coulom, 2007; Browne et al., 2012), integrating three distinct networks: *representation*, *dynamics*, and *prediction*. Specifically, for an observation $x_t$ at time step $t$, the search determines an action $a_{t+1}$ using multiple simulations, each consisting of three phases: selection, expansion, and backup. The selection starts from the hidden state root node $s^0$, selecting child nodes recursively until an unexpanded leaf node $s^l$ is reached. For each non-leaf node $s^k$, the child node $s^{k+1}$ (corresponding to action $a^{k+1}$) is selected according to the highest PUCT (Rosin, 2011; Silver et al., 2017) score:

$$Q(s^k, a^{k+1}) + P(s^k, a^{k+1}) \times \frac{\sqrt{\sum_b N(s^k, b)}}{1 + N(s^k, a^{k+1})} \times c_{puct}, \tag{1}$$

where $Q(s^k, a^{k+1})$ is the estimated Q-value, $P(s^k, a^{k+1})$ is the prior probability, $N(s^k, a^{k+1})$ is the visit counts, and $c_{puct}$ is a constant for exploration. In the expansion phase, to expand the leaf node $s^l$, the dynamics network $g_\theta$ is applied to perform the environmental transition: $s^l, r^l = g_\theta(s^{l-1}, a^l)$, where $r^l$ is the immediate reward. Note that when $l = 0$, the representation network $h_\theta$ is used to initialize the root node: $s^0 = h_\theta(x_t)$. Then, the prediction network $f_\theta$ is applied to evaluate its policy and value: $p^l, v^l = f_\theta(s^l)$, where $p^l$ is used for $P(s^l, a)$ and $v^l$ is the estimated value for $s^l$. The backup phase uses the obtained value $v^l$ to update the statistics $Q(s^k, a^{k+1})$ and $N(s^k, a^{k+1})$:

$$Q(s^k, a^{k+1}) := \frac{N(s^k, a^{k+1}) \times Q(s^k, a^{k+1}) + G^{k+1}}{N(s^k, a^{k+1}) + 1} \text{ and } N(s^k, a^{k+1}) := N(s^k, a^{k+1}) + 1, \tag{2}$$

where $G^{k+1} = \sum_{\tau=0}^{l-k-1} \gamma^\tau r^{k+1+\tau} + \gamma^{l-k} v^l$ is the cumulative reward discounted by a factor $\gamma$.

During training, MuZero continuously performs self-play and optimization. The self-play process collects game trajectories, including $x_t$, $\pi_t$, $a_{t+1}$, $u_{t+1}$, and $z_t$ for all time steps. For each $x_t$, MCTS is conducted to produce the search policy $\pi_t$. Then, an action $a_{t+1} \sim \pi_t$ is applied to the environment, obtaining an immediate reward $u_{t+1}$ and moving forward to the next observation $x_{t+1}$. In addition, $z_t$ is the n-step return. The optimization process updates the networks by sampling records from collected trajectories. For each sampled record $x_t$, the process uses the networks to unroll it for $K$ steps to obtain $s_t^k$ with corresponding $p_t^k$, $v_t^k$, and $r_t^k$ for $0 \le k \le K$, where $s_t^0 = h_\theta(x_t)$ and $s_t^k, r_t^k = g_\theta(s_t^{k-1}, a_{t+k})$ for $k > 0$. Then, all networks are jointly updated using

$$L_t = \sum_{k=0}^{K} l^p(\pi_{t+k}, p_t^k) + \sum_{k=0}^{K} l^v(z_{t+k}, v_t^k) + \sum_{k=1}^{K} l^r(u_{t+k}, r_t^k) + c||\theta||^2, \tag{3}$$

where $l^p$ is the policy loss, $l^v$ is value loss, $l^r$ is the reward loss, and $c||\theta||^2$ is the L2 normalization.

## 4 OptionZero

### 4.1 Option network

*Options* are the generalization of actions to include temporally extended actions, which is applied interchangeably with primitive actions (Sutton et al., 1999; Bacon et al., 2017). In this context, options on Markov decision process (MDP) form a special case of decision problem known as a semi-Markov decision process (SMDP). Given a state $s_t$ at time step $t$ and an option length $L$, we enumerate all possible options, denoted as $o_{t+1} = \{a_{t+1}, a_{t+2}, ..., a_{t+L}\}$, by considering every sequence of primitive actions starting at $s_t$. When executing the option $o_{t+1}$, we obtain a sequence of states and actions $s_t, a_{t+1}, s_{t+1}, a_{t+2}, ..., s_{t+L-1}, a_{t+L}, s_{t+L}$. Ideally, the probability of selecting each option can be calculated by multiplying the probabilities of each primitive action within the option, as illustrated in Figure 1a. For example, when $L = 4$, the probability of option $o_1 = \{a_1, a_2, a_3, a_4\}$ for $s_0$ is $P(a_1) \times P(a_2) \times P(a_3) \times P(a_4) = 0.8^4 = 0.4096$, where $P(a_i)$ is the probability of selecting action $a_i$. A naive approach to obtaining the option probabilities involves using a policy network to evaluate all possible states from $s_t$ to $s_{t+L}$. However, this approach is computationally expensive, and the number of options grows exponentially as the option length $L$ increases, making it infeasible to generate all options.

In practice, since most options occur infrequently due to their lower probabilities, our primary interest lies in the *dominant option*. The dominant option, $o_1 = \{a_1, a_2, ..., a_l\}$, is defined such that

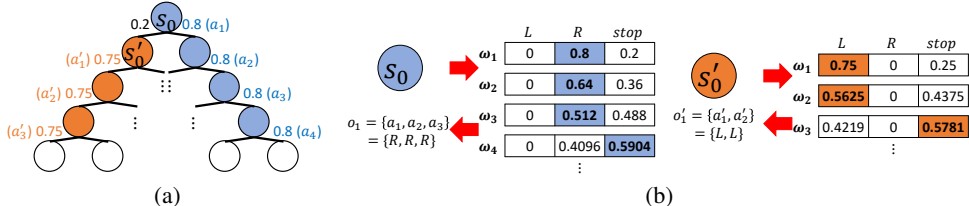

Figure 1: An illustration of calculating option in a decision tree. Each node represents a state with two possible actions, $L$ and $R$, corresponding to the left and right transitions to the subsequent state. (a) The decision tree and probabilities for each option at state $s$. (b) The procedure of determining the dominant option from the option network.

$\Pi_{i=1}^{l} P(a_i) > 0.5 \wedge \Pi_{i=1}^{l+1} P(a_i) \leq 0.5$, where $\Pi_{i=1}^{l} P(a_i)$ is the cumulative product of probabilities and $1 \leq l \leq L$. For example, in Figure 1a, the dominant option for $s_0$ is $o_1 = \{a_1, a_2, a_3\}$ because $P(a_1) \times P(a_2) \times P(a_3) = 0.512$ and $P(a_1) \times P(a_2) \times P(a_3) \times P(a_4) = 0.4096$, and the dominant option for $s'_0$ is $o'_1 = \{a'_1, a'_2\}$. This indicates that the length of the dominant option can vary, depending on how long the cumulative probabilities remain above the threshold of 0.5. In addition, this design ensures that there is only one dominant option for each state $s$, effectively preventing exponential growth in the number of possible options.

Next, we incorporate the *option network* into the prediction network in MuZero, denoted as $\Omega, p, v = f_\theta(s)$, which predicts an additional option output, $\Omega$, for predicting the dominant option at state $s$. Given the maximum option length $L$, the option network produces $L$ distributions, $\Omega = \{\omega_1, \omega_2, ..., \omega_L\}$, which are used to derive the dominant option, $o_1 = \{a_1^*, a_2^*, ..., a_l^*\}$, where $a_i^* = \arg\max_a \omega_i(a)$. Each $\omega_i$ represents the conditional cumulative product probability of selecting a sequence of actions from $a_1^*$ to $a_i^*$, i.e, $\omega_i(a_i^*) = \Pi_{j=1}^{i} P(a_j^*)$. Furthermore, a virtual action, called *stop*, is introduced to provide a termination condition. This *stop* action is the sum of probabilities for all actions except $a^*$, defined as $\omega(stop) = 1 - \omega(a^*)$. To derive the dominant option from $\Omega$, we progressively examine each $\omega_i$ from $\omega_1$ to $\omega_L$, selecting $a_i^*$ as $a_i^* = \arg\max_a \omega_i(a)$ until $i = L$ or $a_i^*$ becomes a *stop* action. We provide an example for obtaining the dominant options for state $s_0$ and $s'_0$, as shown in Figure 1b. This method allows for determining the dominant option at any state $s$ without recurrently evaluating future states, reducing the computational costs.

## 4.2 PLANNING WITH DOMINANT OPTION IN MCTS

This subsection describes the modifications to MCTS implemented in OptionZero to incorporate planning with the dominant option. For simplicity, we will use *option* to represent the *dominant option* in the rest of this paper. The planning generally follows the MuZero but with two modifications, including the network architecture and MCTS. For the network architecture, we add an additional option output to the prediction network, denoted as $\Omega^k, p^k, v^k = f_\theta(s^k)$, where $\Omega^k, p^k$, and $v^k$ are the option distribution, policy distribution, and value at state $s^k$, respectively. Note that we use superscript $s^k$ instead of subscript $s_k$ in this subsection. This is because $s^k$ represents the hidden state, obtained after unrolling $k$ steps by the dynamics network from the initial hidden state $s^0$. In contrast, $s_k$ denotes the actual observed state in the environment at time step $k$. As illustrated in the previous section, we can derive the option $o^k$ from $\Omega^k$. The dynamics network, denoted as $s^{k+l}, r^{k+1,k+l} = g_\theta(s^k, \mathcal{A}^{k+1})$, is modified to predict the next hidden state $s^{k+l}$ and the accumulated discounted reward $r^{k+1,k+l}$ upon executing a composite action $\mathcal{A}^{k+1}$ at $s^k$. The composite action, $\mathcal{A}^{k+1}$, can be either a primitive action $a^{k+1}$ or an option $o^{k+1}$ with the length $l$. The accumulated discounted reward $r^{k+1,k+l}$ is computed as $\sum_{i=1}^{l} \gamma^{i-1} r^{k+i,k+i}$, where $r^{i,i}$ represents the single immediate reward obtained by applying $a^i$ at state $s^{i-1}$. Note that the dynamics network supports unrolling the option directly, eliminating the need to recurrently evaluate each subsequent state from $s^k$ to $s^{k+l}$.

Next, we demonstrate the incorporation of options within MCTS. The search tree retains the structure of the original MCTS but includes edges for options that can bypass multiple nodes directly, as shown in Figure 2. This adaptation integrates options subtly while preserving the internal actions within options, allowing the tree to traverse states using either a primitive action or an option. Each

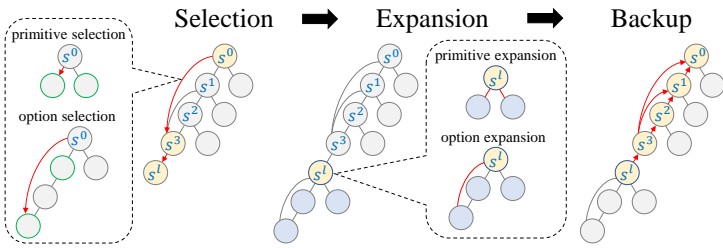

Figure 2: An illustration of each phase in MCTS in OptionZero.

edge within the tree is associated with statistics $\{N(s, \mathcal{A}), Q(s, \mathcal{A}), P(s, \mathcal{A}), R(s, \mathcal{A})\}$, representing its visit counts, estimated Q-value, prior probability, and reward, respectively. Moreover, for nodes that possess both a primitive edge and an option edge, the statistics of the primitive edge are designed to include those of the option edge. For example, if the tree traverses the node via the option edge, the visit counts for both the primitive and option edges are incremented. This ensures the statistics remain consistent with MuZero when only primitive edges are considered within the tree. We illustrate the modifications made to each phase of MCTS in the following.

**Selection.** For any node $s^k$, the selection of the next child node includes two stages: *primitive selection* and *option selection*. The primitive selection only considers primitive child nodes and remains consistent with MuZero by selecting the next action $a^{k+1}$ based on the PUCT score based on equation 1. If the selected action $a^{k+1}$ matches the first action in option $o^{k+1}$, we then proceed with the option selection to determine whether to select this primitive action $a^{k+1}$ or option $o^{k+1}$. Option selection is similar to the primitive selection, using the PUCT score to compare both the primitive and option nodes. Since the option node is a successor node of the primitive node, the statistics for the primitive node need to be adjusted to exclude contributions from the option node in the option selection. We select either primitive or option nodes based on the higher PUCT score, which is calculated as follows:

$$
\begin{cases}
Q(s^k, o^{k+1}) + P(s^k, o^{k+1}) \times \frac{\sqrt{\sum_b N(s^k, b)}}{1 + N(s^k, o^{k+1})} \times c_{puct} & \text{if } \textit{option node}, \\
\tilde{Q}(s^k, a^{k+1}) + \tilde{P}(s^k, a^{k+1}) \times \frac{\sqrt{\sum_b N(s^k, b)}}{1 + (N(s^k, a^{k+1}) - N(s^k, o^{k+1}))} \times c_{puct} & \text{if } \textit{primitive node}.
\end{cases}
\tag{4}
$$

The $\tilde{P}(s^k, a^{k+1}) = \max(0, P(s^k, a^{k+1}) - P(s^k, o^{k+1}))$ ensures that the prior remains non-negative. The adjusted estimated Q-value, $\tilde{Q}(s^k, a^{k+1})$, is calculated as $\frac{N(s^k, a^{k+1})Q(s^k, a^{k+1}) - N(s^k, o^{k+1})Q(s^k, o^{k+1})}{N(s^k, a^{k+1}) - N(s^k, o^{k+1})}$. Note that $\sum_b N(s^k, b)$ is the total visit counts for selecting $a^{k+1}$ and $o^{k+1}$, which is equivalent to $N(s^k, a^{k+1})$ because the statistics of the primitive node already include the statistics of option node. The selection process begins at the root node $s^0$ until an unevaluated node $s^l$ is reached, as shown in Figure 2.

**Expansion.** Assume the last two node in the selection path is $s^m$ and $s^l$, where $s^m$ is the parent node of $s^l$. To expand node $s^l$, we derive $r^{m+1,l}, \Omega^l, p^l, v^l$ using the dynamics and prediction network. The reward $r^{m+1,l}$ is from $s^m$ to $s^l$ and used to initialize the edge $R(s^m, \mathcal{A}^{m+1}) = r^{m+1,l}$. The edge of all primitive child nodes are initialized as $\{N(s^l, a^{l+1}) = R(s^l, a^{l+1}) = Q(s^l, a^{l+1}) = 0\}$ and $P(s^l, a^{l+1}) = p^l$. Then, if the length of option $o^{l+1}$ derived from $\Omega^l$ is larger than 1, we expand the internal nodes following the action within the option. The statistics of each edge are initialized as 0 since these internal nodes are unevaluated. For the option node, the edge is initialized as $\{N(s^l, o^{l+1}) = R(s^l, o^{l+1}) = Q(s^l, o^{l+1}) = 0\}$ and $P(s^l, o^{l+1}) = \omega^l$.

**Backup.** The backup phase updates the visit counts and estimated Q-value from $s^l$ back to $s^0$. Considering that $s^l$ may be accessed through various selection paths from $s^0$, all edges on the possible paths from $s^0$ to $s^l$ must be updated. This ensures that both the visited count and estimated Q-value of all nodes remain consistent within the search, regardless of the selection path chosen. We first obtain the $l - k$-step estimate of the cumulative discounted reward as $G^k = r^{k+1,l} + \gamma^{l-k}v^l$, where $r^{k+1,l}$ is the discounted reward from $s^k$ to $s^l$ and $v^l$ is the value at state $s^l$. Since not all edges have been evaluated, we calculate $r^{k+1,l}$ by using $\frac{r^{1,l} - r^{1,k}}{\gamma^k}$, where $r^{1,k}$ and $r^{1,l}$ represent discounted rewards from the root node $s^0$ to $s^k$ and $s^l$, respectively. Then, we update the estimated Q-value of

each edge, $Q(s^k, \mathcal{A}^{k+1})$, using a similar approach as introduced in equation 2:

$$Q(s^k, \mathcal{A}^{k+1}) := \frac{N(s^k, \mathcal{A}^{k+1}) \times Q(s^k, \mathcal{A}^{k+1}) + G^{k+1}}{N(s^k, \mathcal{A}^{k+1}) + 1},$$

$$N(s^k, \mathcal{A}^{k+1}) := N(s^k, \mathcal{A}^{k+1}) + 1,$$

$$(5)$$

During planning, the MCTS performs a fixed number of simulations, each including the above three phases. Upon the search completed, MCTS selects a child node from the root node $s^0$ based on probabilities proportional to their visit counts and performs the composite action in the environment. Overall, the additional complexity introduced by OptionZero, including the costs for the option network and maintaining statistics for option edges, is negligible compared to the original MuZero.

## 4.3 TRAINING OPTIONZERO

We describe the optimization process using the self-play trajectory in OptionZero, as shown in Figure 3. For better illustration, we utilize three additional symbols, including $\mathcal{O}$, $\mathcal{U}$, $\tau$, and $\hat{\tau}$. Given a state $s_t$ at time step $t$, $\mathcal{O}_i$ represents the $i$-th executed composite action starting from $s_t$, $\mathcal{U}_i$ is defined as the discounted reward obtained after executing $\mathcal{O}_i$, $\tau_i$ denotes the action sequence length of $\mathcal{O}_i$, and $\hat{\tau}_i = \sum_{j=1}^{i} \tau_j$ is the accumulated length from $\mathcal{O}_1$ to $\mathcal{O}_i$. For example, in Figure 3, from the perspective of $s_t$, we can obtain $\mathcal{O}_1 = o_{t+1} = \{a_{t+1}, a_{t+2}\}, \mathcal{O}_2 = \{a_{t+3}\}$, with corresponding discounted rewards $\mathcal{U}_1 = u_{t+1} + \gamma u_{t+2}, \mathcal{U}_2 = u_{t+3}$, action sequence lengths $\tau_1 = 2, \tau_2 = 1$, and accumulated lengths $\hat{\tau}_1 = 2, \hat{\tau}_2 = 3$. Then, the observed discounted reward $\mathcal{U}_1$ at $s_t$ is calculated as $\sum_{i=0}^{\tau_1-1} \gamma^i u_{t+1+i}$, aggregating the observed rewards provided by the environment with a discount factor $\gamma$. The $n$-step return $z_t$ is calculated as $\mathcal{U}_1 + \gamma^{\hat{\tau}_1}\mathcal{U}_2 + ... + \gamma^{\hat{\tau}_{T-1}}\mathcal{U}_T + \gamma^{\hat{\tau}_T} v_{t+\hat{\tau}_T}$, where $\hat{\tau}_T = n$. Note that $v_{t+n}$ is not always available, as $s_{t+n}$ may be bypassed when options are executed. Consequently, we identify the smallest $T$ such that $\hat{\tau}_T \geq n$, ensuring that the step count for the n-step return approximates $n$ as closely as possible. In Figure 3, if $n = 5$, since $s_{t+5}$ is skipped by option, we then approximate the $n$-step return by using $v_{t+6}$ as $z_t = \mathcal{U}_1 + \gamma^2\mathcal{U}_2 + \gamma^3\mathcal{U}_3 + \gamma^4\mathcal{U}_4 + \gamma^6 v_{t+6}$.

Next, we describe the training target for both the policy and option network. The search policy distribution $\pi_t$ is calculated in the same manner as in MuZero. For the option network, given an option length $L$ at state $s_t$, we examine its subsequent states to derive the training target, $\Phi_t = \{\phi_t, \phi_{t+1}, ..., \phi_{t+L-1}\}$. Each $\phi_i$ is a one-hot vector corresponding to the training target for $\omega_i$. Specifically, for any state $s$, if the option network predicts an option $o = \{a_1, a_2, ..., a_l\}$ that exactly matches the composite action $\mathcal{O} = \{a_1, a_2, ..., a_l\}$ executed in the environment, then the option learns the action sequence, i.e., $\phi_i = \text{onehot}(a_{i+1})$ for $0 \leq i \leq l - 1$. Conversely, if $o \neq \mathcal{O}$, then the option learns to *stop*, i.e., $\phi_i = \text{onehot}(stop)$. We iterate this process to set each $\phi$ from $s_t$ to $s_{t+L-1}$. If $\phi_{t+i}$ is set to learn *stop*, subsequent $\phi_{t+j}$ should follow, i.e., $\phi_{t+j} = \text{onehot}(stop)$ for $i \leq j \leq L - 1$. Note that if the length of predicted option $o_i$ is zero, $o_i$ is defined as $\{a_{i+1}\}$, where $a_{i+1} = \arg\max_a p_i(a)$ is determined according to the policy network. This method ensures that the option network eventually learns the cumulative probability of the dominant option, as described in subsection 4.1. Figure 3 shows an example of setting the training target for the option network. If $o_{t+2} \neq \mathcal{O}_2$, then the option network learns $\{a_{t+1}, a_{t+2}, stop\}$, $\{stop, stop, stop\}$, and $\{a_{t+4}, a_{t+5}, a_{t+6}\}$, for $s_t$, $s_{t+2}$, and $s_{t+3}$, respectively.

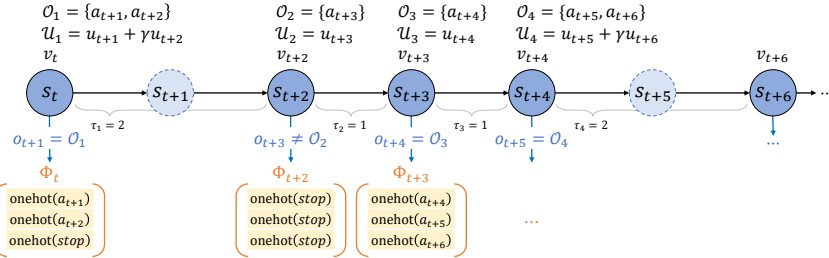

Figure 3: An illustration of optimization in OptionZero. The notion is from the perspective of $s_t$.

During the optimization phase, the sampled state $s_t$ is trained with $K$ unrolling steps, where each step can be either a primitive action or an option. This enables the dynamics network to learn the

environment transitions that incorporate options. The loss is modified from equation 3 as follows:

$$L_t = \sum_{k=0}^{K} l^p(\pi_{t+\hat{\tau}_k}, p_t^k) + \sum_{k=0}^{K} l^v(z_{t+\hat{\tau}_k}, v_t^k) + \sum_{k=1}^{K} l^r(\mathcal{U}_k, r_t^k) + \sum_{k=0}^{K} l^o(\Phi_{t+\hat{\tau}_k}, \Omega_t^k) + c||\theta||^2, \quad (6)$$

where $\hat{\tau}_0 = 0$. Note that the option loss $l^o$ includes $L$ cross-entropy losses.

## 5 EXPERIMENT

### 5.1 OPTIONZERO IN GRIDWORLD

We first train OptionZero in *GridWorld*, a toy environment where the objective is to navigate an agent through a grid map with walls from a start position (S) to a goal (G) via the shortest possible route. The maximum option length is set to nine. Other training details are provided in Appendix A. Figure 4 shows the options learned by the option network at four stages of training: 25%, 50%, 75%, and 100% completion. It can be observed that the learning behavior of OptionZero evolves distinctly across different stages. In the early stage (25%), the model mainly relies on primitive actions, identifying options only when approaching the goal. In the middle stages (50% and 75%), the model begins to establish longer options, progressively learning options with lengths from two up to nine. In the final stage (100%), the model has learned the optimal shortest path using options. Notably, using only primitive actions, the optimal path requires an agent to take at least 30 actions. In contrast, OptionZero achieves this with just four options, accelerating the training process by approximately 7.5 times in this example. This substantial reduction highlights OptionZero's efficacy, especially in more complex environments. This experiment also shows that the option network can progressively learn and refine options during training, without requiring predefined options.

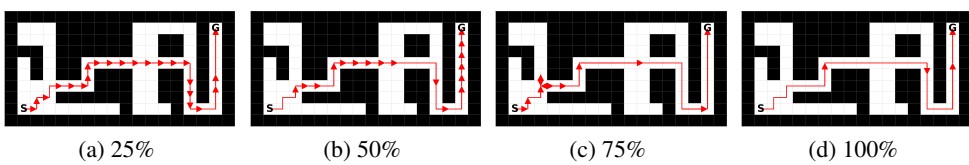

|       (a) 25%       |       (b) 50%       |       (c) 75%       |       (d) 100%       |

Figure 4: Visualization of options learned by OptionZero at different stages of training in GridWorld.

### 5.2 OPTIONZERO IN ATARI GAMES

Next, we evaluate OptionZero on *Atari* games, which are commonly used for investigating options (Sharma et al., 2016; de Waard et al., 2016; Durugkar et al., 2016; Bacon et al., 2017; Vezhnevets et al., 2016; Kim et al., 2023; Lakshminarayanan et al., 2017; Riemer et al., 2020) due to their visually complex environments and subtle frame differences between states, making training with primitive actions inefficient. We train three OptionZero models, denoted as $\ell_1$, $\ell_3$, and $\ell_6$, each configured with maximum option lengths $L = 1, 3$, and 6, respectively. Detailed experiment setups are provided in Appendix B. The model $\ell_1$ serves as a baseline, identical to MuZero, where no options are used during training. In addition, we adopt the frameskip technique (Mnih et al., 2015), commonly used in training on Atari games, set to 4. Namely, this results in a frame difference between 24 states when executing an option of length 6. This requires OptionZero to strategically utilize options when necessary, rather than indiscriminately.

Table 1 shows the results of 26 Atari games. Both $\ell_3$ and $\ell_6$ outperform the baseline $\ell_1$ in mean and median human-normalized scores, with $\ell_3$ achieving the best performance at 1054.30% and 391.69%, representing improvements of 131.58% and 63.29% over $\ell_1$. Overall, 20 out of 26 games perform better than $\ell_1$ for $\ell_3$, and 17 for $\ell_6$. There are 12 games where scores consistently increase as the option length increases. For example, in *up n down*, the scores rise by 63810.83 and 79503.9 from $\ell_1$ to $\ell_6$. Conversely, there are only four games where scores decrease as the option length increases. For example, in *bank heist*, scores drop by 156.54 and 511.26, respectively. We find that this decline is likely due to the difficulty of the dynamics network in learning environment transitions (Vries et al., 2021; He et al., 2024; Guei et al., 2024) in games with more complex action spaces.

As the option length increases, the number of possible option combinations grows. Although we focus on learning the dominant option, the dynamics network still needs to learn across all dominant options. In games like *bank heist*, which offers a wide range of strategic possibilities for different option combinations, the learning complexity for the dynamics network increases. Nevertheless, most of the games still improve when training with options, demonstrating that options enable more effective planning.

Table 1: Scores on 26 Atari games. Bold text in $\ell_3$ and $\ell_6$ indicates scores that surpass $\ell_1$.

| Game | Random | Human | OptionZero | | |
| --- | --- | --- | --- | --- | --- |
| | | | $\ell_1$ | $\ell_3$ | $\ell_6$ |
| alien | 128.30 | 6,371.30 | 2,437.30 | **2,900.07** | **3,748.73** |
| amidar | 11.79 | 1,540.43 | 780.26 | **820.77** | **862.17** |
| assault | 166.95 | 628.89 | 18,389.88 | **19,302.04** | **21,593.53** |
| asterix | 164.50 | 7,536.00 | 177,128.50 | **188,999.00** | **187,716.00** |
| bank heist | 21.70 | 644.50 | 1,097.63 | 950.13 | 906.53 |
| battle zone | 3,560.00 | 33,030.00 | 53,326.67 | **53,583.33** | 39,556.67 |
| boxing | -1.46 | 9.61 | 97.71 | 95.09 | 96.00 |
| breakout | 1.77 | 27.86 | 371.30 | **375.58** | 364.11 |
| chopper command | 644.00 | 8,930.00 | 43,951.67 | **60,181.67** | **45,518.67** |
| crazy climber | 9,337.00 | 32,667.00 | 110,634.00 | **114,390.00** | **128,455.67** |
| demon attack | 208.25 | 3,442.85 | 103,823.17 | **117,270.57** | **109,092.33** |
| freeway | 0.17 | 25.61 | 29.46 | **31.06** | **31.45** |
| frostbite | 90.80 | 4,202.80 | 3,183.40 | **3,641.10** | **6,047.97** |
| gopher | 250.00 | 2,311.00 | 70,985.27 | 68,240.60 | 63,951.47 |
| hero | 1,580.30 | 25,839.40 | 13,568.20 | **19,073.18** | **19,919.22** |
| jamesbond | 33.50 | 368.50 | 8,155.50 | **13,276.67** | **8,571.17** |
| kangaroo | 100.00 | 2,739.00 | 8,929.67 | **12,294.00** | **13,951.33** |
| krull | 1,151.90 | 2,109.10 | 10,255.37 | 10,098.83 | 9,587.57 |
| kung fu master | 304.00 | 20,786.80 | 66,304.67 | **68,528.33** | **69,452.33** |
| ms pacman | 197.80 | 15,375.05 | 3,695.60 | **4,952.37** | **4,706.63** |
| pong | -17.95 | 15.46 | 19.37 | 15.49 | 17.39 |
| private eye | 662.78 | 64,169.07 | 116.83 | 90.76 | 83.24 |
| qbert | 159.38 | 12,085.00 | 17,155.50 | **30,748.42** | **36,328.08** |
| road runner | 200.00 | 6,878.00 | 26,971.33 | **32,786.67** | 21,699.67 |
| seaquest | 215.50 | 40,425.80 | 3,592.53 | **5,606.63** | **6,754.50** |
| up n down | 707.20 | 9,896.10 | 217,021.60 | **280,832.43** | **360,336.33** |
| Mean (%) | 0.00 | 100.00% | 922.72% | **1054.30%** | **1025.56%** |
| Median (%) | 0.00 | 100.00% | 328.40% | **391.69%** | **329.77%** |

## 5.3 OPTION UTILIZATION AND BEHAVIOR ANALYSIS

This subsection analyzes how options are applied to better understand the planning process of OptionZero. Table 2 presents the average percentages of primitive actions (% $a$) and options (% $o$), and the distribution of different option lengths (% $l$) across 26 games. In addition, columns "$\bar{l}$", "% Rpt.", and "% NRpt." show the average action sequence length executed in games, the proportions of options that repeat a single primitive action or involve more than one action types, respectively. Detailed statistics for each game are provided in Appendix D. From the table, we observe that primitive actions are generally the majority, accounting for over 60% in both $\ell_3$ and $\ell_6$. This is because Atari uses a frameskip of four, which means that each primitive action already spans across four states. The use of frameskip four is well-established in previous research, and our experiments further corroborate these findings. However, there are still nearly 30% of states that can adopt options. When comparing the use of options, it is notable that $\ell_6$ applies options less frequently, with a percentage of 30.57% compared to 37.62% in $\ell_3$. However, the average action sequence length for $\ell_6$ (2.03) is longer than that of $\ell_3$ (1.69). This is because action sequences that involve taking two consecutive three-step options in $\ell_3$ are merged into a single six-step option in $\ell_6$, resulting in a lower usage rate of options in statistics. In summary, our findings reveal that OptionZero strategically learns to utilize options as well as employ primitive actions at critical states instead of indiscriminately utilizing longer options.

Next, among the different option lengths used, we observe that generally the longer option lengths are preferred. This suggests that if a state already has applicable options, it is likely these options will be extended further, resulting in a trend towards longer options. This behavior is consistent with the

Table 2: Proportions of options with different lengths and options with repeated primitive actions for $\ell_3$ and $\ell_6$ in Atari games.

| | % a | % o | % 2 | % 3 | % 4 | % 5 | % 6 | $\bar{l}$ | % Rpt. | % NRpt. |
|---|---|---|---|---|---|---|---|---|---|---|
| $\ell_3$ | 62.38% | 37.62% | 6.23% | 31.39% | - | - | - | 1.69 | 75.94% | 24.06% |
| $\ell_6$ | 69.43% | 30.57% | 8.55% | 3.52% | 1.86% | 0.99% | 15.64% | 2.03 | 74.12% | 25.88% |

gradual increase in option lengths observed in gridworld as described in subsection 5.1, illustrating the capability of OptionZero to effectively discover and extend longer options when beneficial.

Finally, we investigate the composition of primitive actions in options. From Table 2, approximately 75% of options consist of repeated primitive actions, similar to the findings in Sharma et al. (2016). For example, in *freeway*, a game where players control chickens across a traffic-filled highway from bottom to top, the most commonly used options by OptionZero are sequences of repeated *Up* actions (*U-U-U* in $\ell_3$ and *U-U-U-U-U-U* in $\ell_6$), guiding the chicken to advance upwards. In addition, OptionZero prefers repeated *Noop* actions, strategically pausing to let cars pass before proceeding. On the other hand, some games still require options composed of diverse combinations of primitive action. For example, in *crazy climber*, a game where players control the left and right side of the body to climb up to the top, OptionZero utilizes options consisting of non-repeated actions. These options often interleave *Up* and *Down* actions to coordinate the movements of the player's hands and feet, respectively. More interestingly, OptionZero also learns to acquire options involving combination skills under specific circumstances. In *hero*, only 4.60% of options involve non-repeated actions. Although the chance is small, such options are crucial during the game. For example, as depicted in Figure 5, the agent executes a series of strategically combined options, including landing from the top, planting a bomb at the corner to destroy a wall, swiftly moving away to avoid injury from the blast, and then skillfully times its movement to the right while firing after the wall is demolished. It is worth noting that there are a total of 24 primitive actions executed from Figure 5a to 5e, but only four options are executed in practice, showing that using option provides effective planning. In conclusion, our results demonstrate that OptionZero is capable of learning complex action sequences tailored to specific game dynamics, effectively discovering the required combinations whether the options are shorter, longer, or contain repeated actions. We have provided the top frequency of options used in each game in the Appendix D.5.

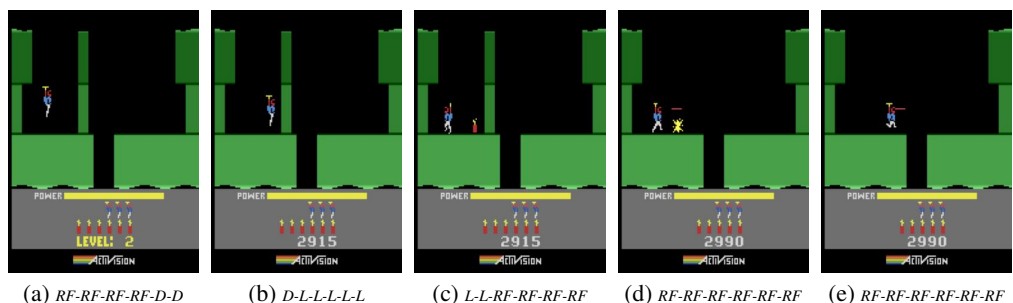

(a) *RF-RF-RF-RF-D-D*    (b) *D-L-L-L-L-L*    (c) *L-L-RF-RF-RF-RF*    (d) *RF-RF-RF-RF-RF-RF*    (e) *RF-RF-RF-RF-RF-RF*

Figure 5: Sequence of game play from (a) to (e) for OptionZero in *hero*. The actions *R*, *L*, *D*, and *F* represent moving right, moving left, placing bombs, and firing, respectively.

## 5.4 OPTION UTILIZATION IN THE SEARCH

We further investigate the options used during planning. Table 3 lists the results for $\ell_1$, $\ell_3$, and $\ell_6$, including the proportions of search trees that consist of at least one option edge is expanded in MCTS ("% in Tree"), the proportions of simulations that at least one option has been selected during search ("% in Sim."), the average tree depth, the median tree depth, and the maximum tree depth. Detailed statistics for each game are provided in Appendix D.3. The results show that approximately 90% of search trees expand options, but only around 30% of search trees choose options during selection. Considering the nature of exploration in MCTS, it is reasonable that not all simulations

will incorporate options. Surprisingly, there are still certain game states for which the search process does not use options at all. Especially in *hero*, From $\ell_3$ to $\ell_6$, the proportion of search trees utilizing options decreases from 74.43% to 54.39%, showing that there are numerous game states where options are not required. However, the performance remains consistent, suggesting that the planning could concentrate on applying options in certain states. Note that the less frequent use of options does not cause undesirable results; eventually, the search behaves similarly to that of MuZero.

Table 3: Proportions of options in search tree for $\ell_3$ and $\ell_6$ in Atari games.

|  | % in Tree | % in Sim. | Avg. tree depth | Median tree depth | Max tree depth |
|---|---|---|---|---|---|
| $\ell_1$ | 0.00% | 0.00% | 14.52 | 12.58 | 48.54 |
| $\ell_3$ | 91.43% | 28.94% | 20.74 | 18.23 | 121.46 |
| $\ell_6$ | 87.48% | 22.28% | 24.92 | 19.35 | 197.58 |

Finally, we compare the tree depths of the MCTS process with and without options. It is naturally considered that applying options provides a deeper tree, which helps in identifying longer future state sequences for better planning and avoiding pitfalls. From the statistics, the average search tree depths generally increase as the maximum option length increases, rising by 6.22 from $\ell_1$ to $\ell_3$ and by 10.4 from $\ell_1$ to $\ell_6$. Interestingly, there are counterexamples where the average depth decreases, such as *hero*. Although the average tree depth decreases in *hero* (22.30, 17.06, and 12.15 for $\ell_1$, $\ell_3$, and $\ell_6$), the performance is improved, as shown in Table 1. Furthermore, by comparing the median tree depth (19, 10, and 7) and maximum tree depth (50, 147, and 276) in *hero*, it can be derived that the model learns to perform deep searches depending on whatever the state requires. Ultimately, whether to conduct a deeper or shallower search tree is learned by OptionZero automatically. For the maximum tree depth, the baseline $\ell_1$ approaches the simulation budget of 50 nodes, meaning the search process may continuously exploit the same promising branch. When integrating with options, although the maximum depths increase, they do not always approach the simulation budgets of 150 and 300. The average numbers of maximum depths are 48.54, 121.46, and 192.27, equivalent to 97.08%, 80.97%, and 64.09% of the budgets, reflecting that the maximum depth is converging. This observation suggests that using an option length of 3 or 6 is sufficient in Atari games.

## 6 DISCUSSION

This paper presents *OptionZero*, a method that integrates options into the well-known MuZero algorithm. OptionZero autonomously discovers options through self-play games and efficiently simulates environment transitions across multiple states with options during planning, which not only eliminates the requirement for obtaining options in advance but also reduces the overhead for examining consecutive states during planning. The empirical results on Atari games demonstrate a significant improvement of 131.58% in mean human-normalized scores, and the behavior analysis reveals that OptionZero effectively discovers options tailored to the specific challenges of each environment. In conclusion, our findings suggest that OptionZero not only discovers options without human knowledge but also maintains efficiency during planning. This makes OptionZero easily applicable to other applications, further extending the versatility of the MuZero algorithm.

As OptionZero builds upon MuZero, it can be easily applied to various environments. For example, when applied to two-player games, OptionZero is expected to discover optimal strategies for both players at specific states. In strategic games such as StarCraft, our approach can learn skillfully combined options, enhancing further explainability and facilitating human learning, as illustrated in subsection 5.3. OptionZero can also be integrated with Sampled MuZero (Hubert et al., 2021) to support environments with complex action spaces, like robotic environments. Nevertheless, our experiments show that OptionZero does not improve performance in all games, especially in environments with numerous option types or visually complex observations, the dynamics network might struggle to learn well. Future work could explore integrating OptionZero with other dynamics models, such as S4 (Gu et al., 2021) or Dreamer (Hafner et al., 2020). Finally, the current design of the option networks requires a predefined maximum option length. Dynamically extending this maximum option length could be a promising direction for future work. We hope our approach and findings provide promising directions in planning with reinforcement learning for future researchers.

ETHICS STATEMENT

We do not foresee any ethical issues in this research work. All data are generated by our programs.

REPRODUCIBILITY STATEMENT

For reproducing the work, we have provided the details of the proposed algorithm in Section 4 and Appendix A, and the setup of training in Appendix B. The source code, scripts for processing behavior analysis, and trained models are available at https://rlg.iis.sinica.edu.tw/papers/optionzero.

ACKNOWLEDGEMENT

This research is partially supported by the National Science and Technology Council (NSTC) of the Republic of China (Taiwan) under Grant Number NSTC 113-2221-E-001-009-MY3, NSTC 113-2634-F-A49-004, and NSTC 113-2221-E-A49-127.

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

## A    IMPLEMENTATION DETAILS

In this section, we detail our OptionZero implementation, which is built upon a publicly available MuZero framework (Wu et al., 2025).

### A.1    MCTS DETAILS

The MCTS implementation mainly follows that introduced in Section 4.2, with minor details described below.

**Dirichlet noise**    To encourage exploration, in MuZero, Dirichlet noise is applied to the root node. Similarly, in OptionZero, since option can also be executed in the environment, we apply Dirichlet noise to both primitive selection and option selection at the root node.

**Default estimated Q value**    For primitive selection, we follow the default estimated Q value for Atari games in the framework (Wu et al., 2025) that enhances exploration:

$$\hat{Q}(s) = \begin{cases} \frac{Q_\Sigma(s)}{N_\Sigma(s)} & N_\Sigma(s) > 0 \\ 1 & N_\Sigma(s) = 0, \end{cases} \tag{7}$$

where $N_\Sigma(s) = \sum_b \mathbf{1}_{N(s,b)>0}$, $Q_\Sigma(s) = \sum_b \mathbf{1}_{N(s,b)>0} Q(s,b)$, and $\mathbf{1}_{N(s,b)>0}$ is the characteristic function that only considers primitive child nodes with non-zero visit counts.

For option selection, since the contributions of option child node are included in the statistics of its corresponding predecessor primitive child node, we use a default estimated Q value that incorporates a virtually losing outcome:

$$\hat{Q}(s) = \frac{Q_{(s,a)} \times N(s,a)}{N(s,a) + 1}, \tag{8}$$

where $N(s,a)$ is the visit counts of the primitive child node, and $Q_{(s,a)}$ is the mean value of the primitive child node.

### A.2    MCTS COMPLEXITY

The complexity of the modified MCTS remains the same as the original, with additional minor computational costs in introducing a new network head to predict and use the dominant option. Specifically, in the selection phase, the only added step is comparing the PUCT scores of option child nodes and primitive child nodes, as in equation 4. In the expansion phase, the option policy $\Omega$ is evaluated along with policy $p$ and value $v$. Since most network weights of the option policy head are shared with the rest of the network, the impact on runtime is negligible. While more nodes are initially expanded, each simulation evaluates only one node at a time. In the backup phase, as the statistics of option edges can be easily derived from primitive edges, only the statistics of primitive edges are maintained in practice, eliminating the additional cost of updating all possible option edges.

### A.3    GRIDWORLD ENVIRONMENT

The implementation is also built upon the same framework (Wu et al., 2025), with a custom Grid-World environment added. The reward of the environment is defined as follows: the initial total reward is 200 points, and for each action or option taken, one point is deducted from the reward.

### A.4    NETWORK ARCHITECTURE

The network architecture follows a structure similar to MuZero. As discussed in Section 4, the option network is incorporated into the prediction network. Specifically, besides the policy head, we add additional $L - 1$ option heads for predicting $\Omega = \{\omega_2, \omega_3, ..., \omega_L\}$, initialized to predict the *stop*. Note that there is no need for extra prediction of $\omega_1$, since we can directly get the first action of the dominant option from policy head by choosing $a_1^* = \arg\max_a p(a)$. Additionally, the dynamics network is modified to simulate the environment transitions of executing both primitive actions and

options. By extending the original action input, the action sequence input to the dynamics network is encoded into a fixed number of $L$ planes for supporting options with different lengths, with the $l$th plane corresponding to the $l$th move inside an option. Note that when $l < L$, the subsequent planes are set to zero, representing no moves.

**Atari games**   We additionally adopt the state consistency (Ye et al., 2021). Therefore, the SimSiam (Chen & He, 2021) architecture is included to calculate the consistency loss.

**GridWorld**   The network architecture generally follows the architecture tailored for Atari games. However, in the design of the representation network, we removed the down-sampling mechanism, adopting a setup similar to MuZero for board games as in Wu et al. (2025).

## B   TRAINING OPTIONZERO

In this section, we describe the details for training OptionZero models used in the experiments. The experiments are conducted on machines with 24 CPU cores and four NVIDIA GTX 1080 Ti GPUs. For the training configurations, we generally follow those in MuZero, where the hyperparameters are listed in Table 4.

Table 4: Hyperparameters for training.

| Parameter | Atari | Grid world |
|---|---|---|
| Optimizer | SGD | |
| Optimizer: learning rate | 0.1 | |
| Optimizer: momentum | 0.9 | |
| Optimizer: weight decay | 0.0001 | |
| Discount factor | 0.997 | |
| Priority exponent ($\alpha$) | 1 | |
| Priority correction ($\beta$) | 0.4 | |
| Bootstrap step (n-step return) | 5 | |
| MCTS simulation | 50 | |
| Softmax temperature | 1 | |
| Frames skip | 4 | - |
| Frames stacked | 4 | - |
| Iteration | 300 | 400 |
| Training steps | 60k | 80k |
| Batch size | 512 | 1024 |
| # Blocks | 2 | 1 |
| Replay buffer size | 1M frames | 8k games |
| Max frames per episode | 108k | - |
| Dirichlet noise ratio | 0.25 | 0.3 |

**Atari games**   For Atari games, each setting is trained for 3 runs on each game, with each model taking approximately 22 hours to complete. Since we introduce an additional head to predict option, the training time slightly increases as the max option length increases. For $\ell_1$, the training time is approximately 21.89 hours. For $\ell_3$ and $\ell_6$, the training times increase to around 22.28 hours and 22.95 hours, representing increases of 1.8% and 4.8%, respectively. The performance is measured based on the average score of the latest 100 completed games in each run during the training (Hessel et al., 2021). The training curves are shown in Figure 6.

**GridWorld**   In this toy example, we aim to clearly show that the length of the learned options can be extended as the training time increases. For training, we use a maximum option length $L = 9$, fixing the goal position and selecting random starting points. As for the evaluation, we fix the starting point and the goal as shown in Figure 4. The evaluation also uses 50 MCTS simulations, and the original softmax function is replaced with max selection.

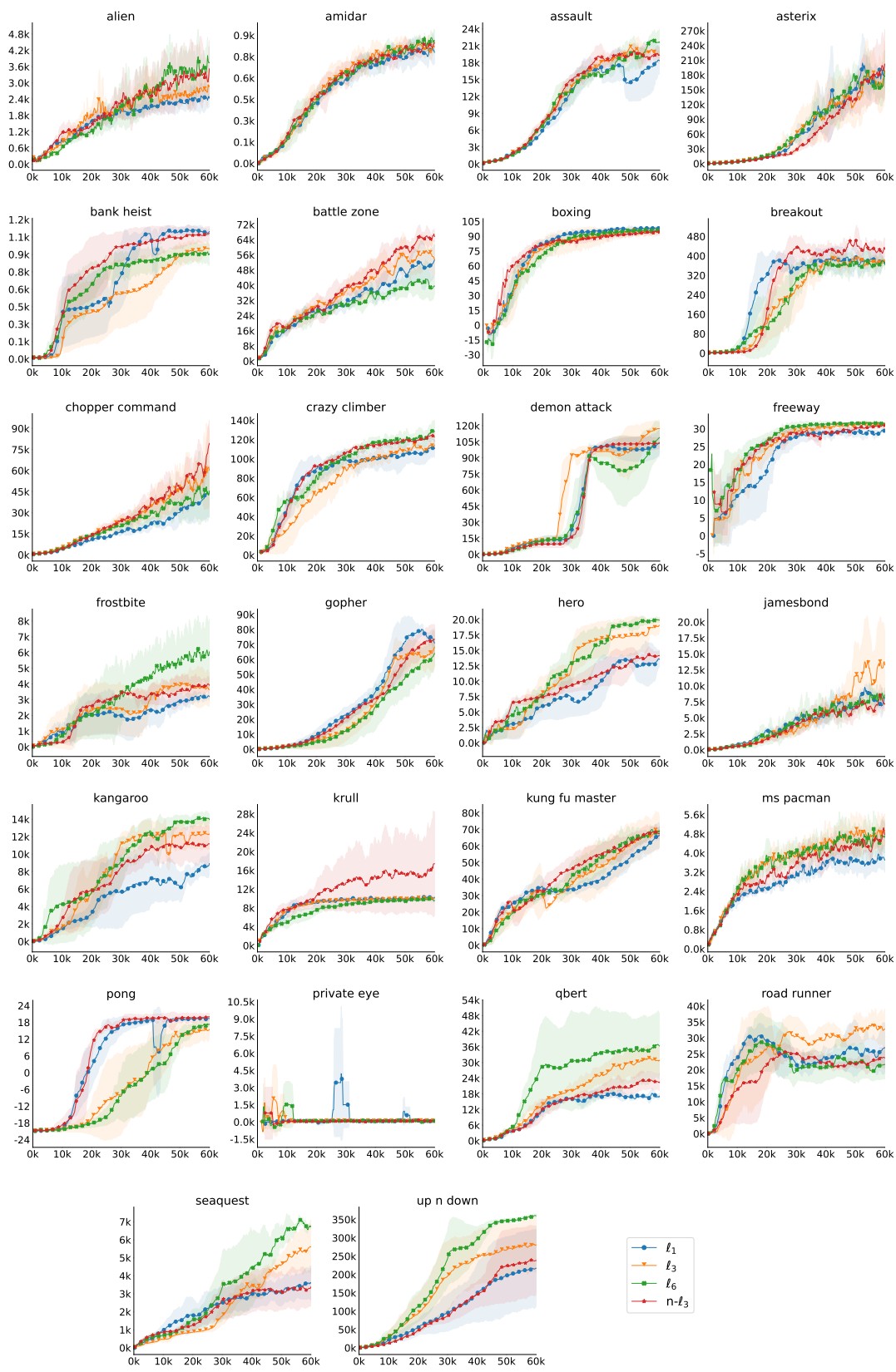

Figure 6: Training curves on 26 Atari games.

# C  ABLATION STUDY FOR OPTIONZERO

For the ablation study, we train OptionZero with a maximum option length $L = 3$, but disable the execution of options in the environment, using them solely for MCTS planning, denoted as n-$\ell_3$. Since all composite actions are primitive actions, we define each $o_i$ as $\{a_{i+1}\}$, where $a_{i+1} = \arg\max_a p_i(a)$ and train the option network according to the policy network. According to the results shown in Table 5, n-$\ell_3$ achieves a mean human-normalized score of 1008.15%, which is 85.43% higher than the baseline $\ell_1$, indicating that OptionZero still enhances MCTS planning with options without executing them. Notably, in games that require precise step-by-step predictions, such as *gopher*, n-$\ell_3$ outperforms $\ell_3$, indicating that planning for every step remains crucial for certain games. However, in games that benefit from bypassing unimportant frames, such as *seaquest*, the performance of n-$\ell_3$ is only comparable to baseline.

Table 5: Scores on 26 Atari games for the ablation study. Bold text in $\ell_3$ and n-$\ell_3$ indicates scores that surpass $\ell_1$.

| Game | Random | Human | OptionZero $\ell_1$ | OptionZero $\ell_3$ | OptionZero n-$\ell_3$ |
|---|---|---|---|---|---|
| alien | 128.30 | 6,371.30 | 2,437.30 | **2,900.07** | **3,523.20** |
| amidar | 11.79 | 1,540.43 | 780.26 | **820.77** | **848.97** |
| assault | 166.95 | 628.89 | 18,389.88 | **19,302.04** | **19,378.79** |
| asterix | 164.50 | 7,536.00 | 177,128.50 | **188,999.00** | **202,183.33** |
| bank heist | 21.70 | 644.50 | 1,097.63 | 950.13 | 1,081.10 |
| battle zone | 3,560.00 | 33,030.00 | 53,326.67 | **53,583.33** | **65,660.00** |
| boxing | -1.46 | 9.61 | 97.71 | 95.09 | 94.58 |
| breakout | 1.77 | 27.86 | 371.30 | **375.58** | **427.18** |
| chopper command | 644.00 | 8,930.00 | 43,951.67 | **60,181.67** | **79,340.33** |
| crazy climber | 9,337.00 | 32,667.00 | 110,634.00 | **114,390.00** | **122,865.67** |
| demon attack | 208.25 | 3,442.85 | 103,823.17 | **117,270.57** | **104,351.00** |
| freeway | 0.17 | 25.61 | 29.46 | **31.06** | **30.93** |
| frostbite | 90.80 | 4,202.80 | 3,183.40 | **3,641.10** | **3,923.63** |
| gopher | 250.00 | 2,311.00 | 70,985.27 | 68,240.60 | **73,338.67** |
| hero | 1,580.30 | 25,839.40 | 13,568.20 | **19,073.18** | **14,181.65** |
| jamesbond | 33.50 | 368.50 | 8,155.50 | **13,276.67** | 7,172.17 |
| kangaroo | 100.00 | 2,739.00 | 8,929.67 | **12,294.00** | **11,175.33** |
| krull | 1,151.90 | 2,109.10 | 10,255.37 | 10,098.83 | **17,420.13** |
| kung fu master | 304.00 | 20,786.80 | 66,304.67 | **68,528.33** | **67,735.67** |
| ms pacman | 197.80 | 15,375.05 | 3,695.60 | **4,952.37** | **4,762.23** |
| pong | -17.95 | 15.46 | 19.37 | 15.49 | **20.01** |
| private eye | 662.78 | 64,169.07 | 116.83 | 90.76 | 94.71 |
| qbert | 159.38 | 12,085.00 | 17,155.50 | **30,748.42** | **22,321.75** |
| road runner | 200.00 | 6,878.00 | 26,971.33 | **32,786.67** | 23,784.67 |
| seaquest | 215.50 | 40,425.80 | 3,592.53 | **5,606.63** | 3,378.60 |
| up n down | 707.20 | 9,896.10 | 217,021.60 | **280,832.43** | **238,409.40** |
| Normalized Mean | 0.00 | 100.00 % | 922.72 % | **1054.30 %** | **1008.15 %** |
| Normalized Median | 0.00 | 100.00 % | 328.40 % | **391.69 %** | **341.19 %** |

# D   IN-DEPTH BEHAVIOR ANALYSIS

In this experiment section, we conduct detailed analysis for $\ell_3$ and $\ell_6$ in 26 Atari games.

## D.1   OPTIONS APPLIED IN GAMES

We present the statistics in all 26 Atari games conducted for the behavior analysis in Section 5.3. Specifically, we provide the numbers of options types, option usages, proportions of options with repeated actions, and the average option lengths for $\ell_3$ and $\ell_6$ in Table 6 and Table 7, respectively. The columns "# $\{a\}$" and "# $\{o\}$" show the numbers of available primitive actions and the numbers of the options recorded during evaluation, columns "% $a$" and "% $o$" show the proportions of actions and options applied during the game, columns "% Rpt." and "% NRpt." show the proportions of options with repeated primitive actions and options with more than one action type, and column "$\bar{l}$" shows the average options length (including primitive action). We also provide the proportions of options with different lengths for both $\ell_3$ and $\ell_6$ in Table 8.

Table 6: Numbers of option types, option usages, proportions of options with repeated actions, and average option lengths for $\ell_3$ in 26 Atari games.

| Game | # $\{a\}$ | # $\{o\}$ | % $a$ | % $o$ | % Rpt. | % NRpt. | $\bar{l}$ |
|---|---|---|---|---|---|---|---|
| alien | 18 | 185 | 67.66% | 32.34% | 94.54% | 5.46% | 1.61 |
| amidar | 10 | 187 | 65.51% | 34.49% | 98.43% | 1.57% | 1.65 |
| assault | 7 | 139 | 78.84% | 21.16% | 57.20% | 42.80% | 1.30 |
| asterix | 9 | 163 | 92.82% | 7.18% | 90.74% | 9.26% | 1.10 |
| bank heist | 18 | 138 | 41.93% | 58.07% | 7.60% | 92.40% | 2.05 |
| battle zone | 18 | 217 | 88.89% | 11.11% | 95.33% | 4.67% | 1.18 |
| boxing | 18 | 240 | 39.53% | 60.47% | 50.54% | 49.46% | 2.12 |
| breakout | 4 | 64 | 79.00% | 21.00% | 84.03% | 15.97% | 1.32 |
| chopper command | 18 | 242 | 76.08% | 23.92% | 92.28% | 7.72% | 1.41 |
| crazy climber | 9 | 158 | 50.93% | 49.07% | 25.09% | 74.91% | 1.94 |
| demon attack | 6 | 153 | 76.70% | 23.30% | 88.74% | 11.26% | 1.38 |
| freeway | 3 | 30 | 39.20% | 60.80% | 95.04% | 4.96% | 2.19 |
| frostbite | 18 | 273 | 58.10% | 41.90% | 94.14% | 5.86% | 1.81 |
| gopher | 8 | 325 | 51.44% | 48.56% | 68.60% | 31.40% | 1.88 |
| hero | 18 | 346 | 81.72% | 18.28% | 95.40% | 4.60% | 1.34 |
| jamesbond | 18 | 376 | 51.34% | 48.66% | 67.24% | 32.76% | 1.90 |
| kangaroo | 18 | 230 | 29.84% | 70.16% | 70.54% | 29.46% | 2.36 |
| krull | 18 | 182 | 67.72% | 32.28% | 54.58% | 45.42% | 1.54 |
| kung fu master | 14 | 536 | 38.48% | 61.52% | 70.51% | 29.49% | 2.15 |
| ms pacman | 9 | 181 | 65.01% | 34.99% | 94.70% | 5.30% | 1.67 |
| pong | 6 | 159 | 24.57% | 75.43% | 76.98% | 23.02% | 2.47 |
| private eye | 18 | 233 | 93.60% | 6.40% | 78.74% | 21.26% | 1.08 |
| qbert | 6 | 105 | 50.04% | 49.96% | 97.84% | 2.16% | 1.97 |
| road runner | 18 | 144 | 85.95% | 14.05% | 65.92% | 34.08% | 1.23 |
| seaquest | 18 | 340 | 74.88% | 25.12% | 70.92% | 29.08% | 1.38 |
| up n down | 6 | 116 | 52.06% | 47.94% | 88.89% | 11.11% | 1.91 |
| Average | - | - | 62.38% | 37.62% | 75.94% | 24.06% | 1.69 |

Table 7: Numbers of option types, option usages, proportions of options with repeated actions, and average option lengths for $\ell_6$ in 26 Atari games.

| Game | #$\{a\}$ | #$\{o\}$ | %$a$ | %$o$ | % Rpt. | % NRpt. | $\bar{l}$ |
|---|---|---|---|---|---|---|---|
| alien | 18 | 411 | 69.28% | 30.72% | 94.13% | 5.87% | 2.30 |
| amidar | 10 | 318 | 67.62% | 32.38% | 97.19% | 2.81% | 2.22 |
| assault | 7 | 367 | 78.71% | 21.29% | 58.78% | 41.22% | 1.35 |
| asterix | 9 | 199 | 92.68% | 7.32% | 86.80% | 13.20% | 1.10 |
| bank heist | 18 | 588 | 48.17% | 51.83% | 11.88% | 88.12% | 2.28 |
| battle zone | 18 | 513 | 91.54% | 8.46% | 95.21% | 4.79% | 1.24 |
| boxing | 18 | 568 | 48.46% | 51.54% | 40.29% | 59.71% | 2.77 |
| breakout | 4 | 132 | 81.21% | 18.79% | 85.92% | 14.08% | 1.28 |
| chopper command | 18 | 351 | 82.75% | 17.25% | 87.56% | 12.44% | 1.40 |
| crazy climber | 9 | 724 | 60.69% | 39.31% | 26.15% | 73.85% | 2.58 |
| demon attack | 6 | 301 | 82.64% | 17.36% | 88.16% | 11.84% | 1.34 |
| freeway | 3 | 118 | 45.38% | 54.62% | 90.66% | 9.34% | 3.45 |
| frostbite | 18 | 708 | 66.80% | 33.20% | 86.23% | 13.77% | 2.45 |
| gopher | 8 | 692 | 56.12% | 43.88% | 64.71% | 35.29% | 2.01 |
| hero | 18 | 576 | 89.58% | 10.42% | 85.05% | 14.95% | 1.39 |
| jamesbond | 18 | 735 | 66.08% | 33.92% | 86.88% | 13.12% | 2.30 |
| kangaroo | 18 | 718 | 40.00% | 60.00% | 64.44% | 35.56% | 3.43 |
| krull | 18 | 679 | 60.31% | 39.69% | 45.89% | 54.11% | 2.07 |
| kung fu master | 14 | 1386 | 53.09% | 46.91% | 53.40% | 46.60% | 2.53 |
| ms pacman | 9 | 219 | 77.13% | 22.87% | 94.95% | 5.05% | 1.77 |
| pong | 6 | 741 | 36.82% | 63.18% | 61.09% | 38.91% | 3.71 |
| private eye | 18 | 488 | 97.04% | 2.96% | 77.05% | 22.95% | 1.06 |
| qbert | 6 | 450 | 62.79% | 37.21% | 92.84% | 7.16% | 2.52 |
| road runner | 18 | 225 | 96.19% | 3.81% | 90.55% | 9.45% | 1.10 |
| seaquest | 18 | 621 | 82.41% | 17.59% | 76.38% | 23.62% | 1.38 |
| up n down | 6 | 226 | 71.65% | 28.35% | 84.88% | 15.12% | 1.84 |
| Average | - | - | 69.43% | 30.57% | 74.12% | 25.88% | 2.03 |

Table 8: Proportions of options with different lengths for $\ell_3$ and $\ell_6$ in 26 Atari games.

| Game | $\ell_3$ | | | $\ell_6$ | | | | | |
|---|---|---|---|---|---|---|---|---|---|
| | %1 | %2 | %3 | %1 | %2 | %3 | %4 | %5 | %6 |
| alien | 67.66% | 4.03% | 28.31% | 69.28% | 4.24% | 1.47% | 0.97% | 0.60% | 23.44% |
| amidar | 65.51% | 3.52% | 30.97% | 67.62% | 7.09% | 2.36% | 1.71% | 0.60% | 20.61% |
| assault | 78.84% | 11.85% | 9.30% | 78.71% | 11.74% | 6.61% | 1.90% | 0.63% | 0.42% |
| asterix | 92.82% | 4.02% | 3.17% | 92.68% | 5.99% | 0.62% | 0.29% | 0.13% | 0.28% |
| bank heist | 41.93% | 11.46% | 46.62% | 48.17% | 20.37% | 10.52% | 6.66% | 4.31% | 9.96% |
| battle zone | 88.89% | 4.34% | 6.77% | 91.54% | 3.16% | 1.33% | 0.53% | 0.22% | 3.23% |
| boxing | 39.53% | 8.56% | 51.91% | 48.46% | 13.74% | 6.25% | 2.73% | 1.77% | 27.05% |
| breakout | 79.00% | 9.96% | 11.04% | 81.21% | 13.30% | 3.81% | 0.77% | 0.29% | 0.63% |
| chopper command | 76.08% | 7.09% | 16.83% | 82.75% | 9.00% | 2.16% | 1.08% | 1.32% | 3.70% |
| crazy climber | 50.93% | 4.63% | 44.44% | 60.69% | 6.37% | 3.30% | 1.44% | 0.71% | 27.49% |
| demon attack | 76.70% | 8.23% | 15.07% | 82.64% | 9.64% | 3.46% | 1.70% | 0.69% | 1.87% |
| freeway | 39.20% | 2.61% | 58.19% | 45.38% | 4.93% | 1.91% | 0.98% | 0.50% | 46.29% |
| frostbite | 58.10% | 2.83% | 39.08% | 66.80% | 3.29% | 1.67% | 0.92% | 0.63% | 26.70% |
| gopher | 51.44% | 9.24% | 39.32% | 56.12% | 22.73% | 6.51% | 3.45% | 1.56% | 9.64% |
| hero | 81.72% | 2.46% | 15.82% | 89.58% | 2.57% | 0.66% | 0.24% | 0.17% | 6.78% |
| jamesbond | 51.34% | 7.10% | 41.55% | 66.08% | 6.68% | 3.14% | 1.27% | 0.99% | 21.83% |
| kangaroo | 29.84% | 4.47% | 65.68% | 40.00% | 8.47% | 4.77% | 3.47% | 1.51% | 41.78% |
| krull | 67.72% | 10.98% | 21.30% | 60.31% | 12.44% | 9.58% | 5.16% | 2.45% | 10.06% |
| kung fu master | 38.48% | 7.74% | 53.77% | 53.09% | 14.46% | 5.03% | 3.53% | 1.60% | 22.28% |
| ms pacman | 65.01% | 2.93% | 32.07% | 77.13% | 7.24% | 2.22% | 0.86% | 0.25% | 12.31% |
| pong | 24.57% | 3.74% | 71.69% | 36.82% | 6.98% | 3.08% | 3.04% | 1.84% | 48.24% |
| private eye | 93.60% | 4.67% | 1.73% | 97.04% | 1.84% | 0.45% | 0.11% | 0.07% | 0.49% |
| qbert | 50.04% | 2.62% | 47.34% | 62.79% | 4.75% | 3.52% | 1.62% | 0.82% | 26.50% |
| road runner | 85.95% | 5.41% | 8.64% | 96.19% | 1.69% | 0.49% | 0.27% | 0.13% | 1.23% |
| seaquest | 74.88% | 12.27% | 12.85% | 82.41% | 9.62% | 2.79% | 1.07% | 0.66% | 3.45% |
| up n down | 52.06% | 5.16% | 42.77% | 71.65% | 9.96% | 3.85% | 2.70% | 1.33% | 10.50% |
| Average | 62.38% | 6.23% | 31.39% | 69.43% | 8.55% | 3.52% | 1.86% | 0.99% | 15.64% |

In addition, we observe other examples of using options in games, which are introduced as follows. The sample videos of 26 games are provided in the supplementary material, in which, for each frame, the video prints the applied move (can be either action or option); additionally, if the planning suggests an option, it is printed below the applied move.

***kung fu master*** There are relatively more options with non-repeated actions since the player requires a combination of *D* and *F*, such as *DR-DR-DRF* in $\ell_3$ to attack the opponents with an uppercut.[1] Notably, the proportions of options with non-repeated actions significantly increased in $\ell_6$, in which the player can even combine two uppercuts in an option, such as *DLF-DR-DR-DLF-DR-DR*.

***ms pacman*** Many options contain only repeated actions, e.g., *UL-UL-UL*.[2] Such a repetition is discovered since the game supports multi-direction control, allowing the agent to simply use the repeated *UL* to go through an L-shape passage without requiring options such as *L-U-U*.

***seaquest*** The game also supports multi-direction control, allowing the agent to use *URF-URF* for moving with firing simultaneously.[3]

---

[1]*L* and *R* for turning left and right, *D* for squatting down, and *F* for attacking.

[2]*U*, *D*, *L*, and *R* for moving up, down, left, and right in the top view, respectively.

[3]*U* and *R* for moving up and right, *F* for firing.

### D.2 LEARNING PROCESS OF OPTIONS

In this experiment section, we show an example to illustrate how longer options are discovered during the training process, using an example of options involving only *R* in *ms pacman*, shown in Figure 7. In the 1st iteration, there are only primitive actions, where *R* is the most frequently used (only 9 primitive actions). Therefore, the agent begins using *R* with increased frequency, and the usage even exceeds 80% in the 3rd iteration. Meanwhile, due to the high usage, the model starts learning options involving more *R*. The options *R-R* and *R-R-R* are therefore becoming the majority in the 4th and the 6th iteration, respectively. Eventually, the agent explores other actions, thus making the *R*-related options suddenly decrease after the 7th iteration. Note that in different training trials, as the agent initially explores randomly, the first option learned does not consistently involve *R*. Depending on how the agent explores the primitive actions, options involving various combinations of actions are discovered at different stages of the training process.

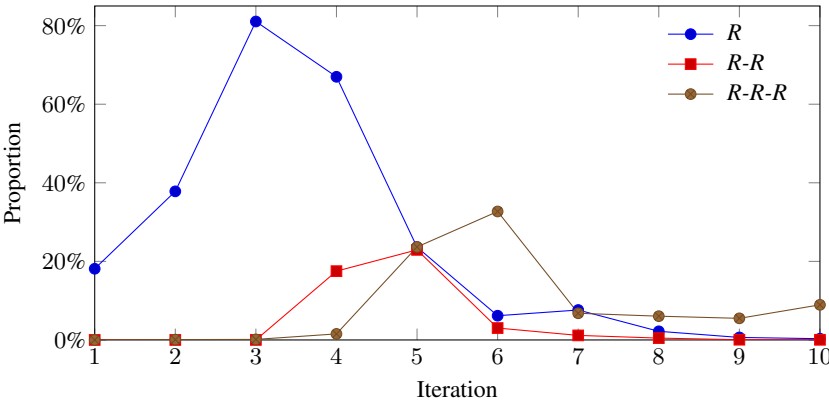

Figure 7: Distribution of options *R*, *R-R*, and *R-R-R* at the beginning of learning *ms pacman*.

### D.3 OPTIONS IN THE SEARCH TREE

In this experiment section, we present the detailed statistics in all 26 Atari games conducted for the behavior analysis in Section 5.4. Specifically, we provide the detailed proportions of options inside the search in Table 9, and the detailed tree depths in Table 10. In Table 9, the columns "% Env." and "% MCTS" represent the proportions of options applied to the environments and the options suggested by the MCTS process. Note that due to softmax selection, the options suggested by the tree, "% MCTS", are not always applied, resulting in a lower "% Env.". Nevertheless, if the suggested options have higher visit counts, they are more likely to be applied, such as in *crazy climber*, *freeway*, and *kung fu master*, where options are well-trained for specific purposes, like climbing, crossing the road, and attacking with an uppercut, respectively, and therefore have higher probabilities of application. Also, the columns "% in Tree" and "% in Sim." represent the proportions of the search tree that contains at least one option (in its 50 simulations) and the proportions of simulations whose selection path contains an option. In Table 10, the columns "Avg" and "Max" represent the average and the maximum search tree depths; the columns "$P_{25}$", "$P_{50}$", and "$P_{75}$" represent the 25th, 50th, and 75th percentiles. Based on the $P_{50}$ and $P_{75}$ depths, we observe that the search is not deep in most cases, implying that the models learn to perform deep searches only in certain states.

Table 9: Proportions of options in search tree for $\ell_3$ and $\ell_6$ in 26 Atari games.

| Game | $\ell_3$ | | | | $\ell_6$ | | | |
|---|---|---|---|---|---|---|---|---|
| | % Env. | % MCTS | % in Tree | % in Sim. | % Env. | % MCTS | % in Tree | % in Sim. |
| alien | 32.34% | 52.05% | 99.81% | 35.85% | 30.72% | 54.92% | 96.80% | 24.13% |
| amidar | 34.49% | 52.26% | 90.68% | 31.95% | 32.38% | 48.54% | 90.17% | 26.82% |
| assault | 21.16% | 33.99% | 82.15% | 11.93% | 21.29% | 35.45% | 90.45% | 11.95% |
| asterix | 7.18% | 15.02% | 63.52% | 7.11% | 7.32% | 17.19% | 51.50% | 4.99% |
| bank heist | 58.07% | 74.56% | 99.45% | 33.10% | 51.83% | 73.85% | 97.71% | 22.65% |
| battle zone | 11.11% | 30.41% | 83.91% | 14.91% | 8.46% | 26.67% | 80.12% | 9.89% |
| boxing | 60.47% | 77.80% | 99.09% | 40.69% | 51.54% | 73.50% | 98.70% | 33.12% |
| breakout | 21.00% | 43.77% | 90.74% | 15.52% | 18.79% | 49.61% | 90.37% | 13.65% |
| chopper command | 23.92% | 41.11% | 88.99% | 21.96% | 17.25% | 33.43% | 90.46% | 15.90% |
| crazy climber | 49.07% | 67.94% | 98.69% | 39.96% | 39.31% | 59.56% | 98.53% | 37.09% |
| demon attack | 23.30% | 36.21% | 93.16% | 16.04% | 17.36% | 29.88% | 82.75% | 11.65% |
| freeway | 60.80% | 86.08% | 99.11% | 43.83% | 54.62% | 74.97% | 98.75% | 38.75% |
| frostbite | 41.90% | 65.81% | 97.58% | 41.18% | 33.20% | 61.49% | 96.33% | 30.52% |
| gopher | 48.56% | 63.83% | 99.17% | 25.33% | 43.88% | 65.21% | 99.37% | 20.31% |
| hero | 18.28% | 39.68% | 74.43% | 20.07% | 10.42% | 26.64% | 62.88% | 11.60% |
| jamesbond | 48.66% | 67.76% | 99.16% | 39.10% | 33.92% | 52.62% | 97.87% | 27.33% |
| kangaroo | 70.16% | 86.86% | 99.64% | 55.49% | 60.00% | 78.64% | 99.69% | 44.32% |
| krull | 32.28% | 54.00% | 99.04% | 27.34% | 39.69% | 69.86% | 99.79% | 28.64% |
| kung fu master | 61.52% | 80.42% | 99.62% | 41.06% | 46.91% | 68.90% | 99.80% | 30.53% |
| ms pacman | 34.99% | 56.34% | 96.90% | 30.82% | 22.87% | 39.34% | 95.10% | 21.84% |
| pong | 75.43% | 86.98% | 99.46% | 54.27% | 63.18% | 77.40% | 99.48% | 43.50% |
| private eye | 6.40% | 27.84% | 67.01% | 7.61% | 2.96% | 14.88% | 44.25% | 5.44% |
| qbert | 49.96% | 72.41% | 99.16% | 38.83% | 37.21% | 59.20% | 93.98% | 29.35% |
| road runner | 14.05% | 28.43% | 62.98% | 12.36% | 3.81% | 8.70% | 33.76% | 3.99% |
| seaquest | 25.12% | 44.43% | 95.88% | 16.88% | 17.59% | 31.77% | 92.72% | 13.66% |
| up n down | 47.94% | 66.05% | 97.74% | 29.12% | 28.35% | 44.52% | 93.03% | 17.53% |
| Average | 37.62% | 55.85% | 91.43% | 28.94% | 30.57% | 49.11% | 87.48% | 22.28% |

Table 10: Tree depths for $\ell_1$, $\ell_3$, and $\ell_6$ in 26 Atari games.

| Game | $\ell_1$ | | | | | $\ell_3$ | | | | | $\ell_6$ | | | | |
|---|---|---|---|---|---|---|---|---|---|---|---|---|---|---|---|
| | Avg | $P_{25}$ | $P_{50}$ | $P_{75}$ | Max | Avg | $P_{25}$ | $P_{50}$ | $P_{75}$ | Max | Avg | $P_{25}$ | $P_{50}$ | $P_{75}$ | Max |
| alien | 17.99 | 10 | 14 | 26 | 50 | 24.19 | 13 | 20 | 31 | 123 | 29.61 | 13 | 22 | 42 | 259 |
| amidar | 13.17 | 8 | 11 | 16 | 49 | 22.62 | 10 | 20 | 32 | 135 | 31.43 | 10 | 22 | 43 | 276 |
| assault | 9.86 | 7 | 9 | 12 | 45 | 8.81 | 6 | 8 | 11 | 78 | 9.30 | 7 | 9 | 11 | 127 |
| asterix | 10.39 | 6 | 7 | 11 | 49 | 7.83 | 5 | 7 | 9 | 100 | 7.15 | 5 | 6 | 8 | 102 |
| bank heist | 13.77 | 12 | 13 | 15 | 49 | 17.10 | 14 | 17 | 20 | 88 | 16.22 | 12 | 15 | 19 | 114 |
| battle zone | 16.70 | 8 | 13 | 24 | 49 | 11.36 | 6 | 9 | 13 | 105 | 10.93 | 6 | 8 | 12 | 144 |
| boxing | 15.13 | 11 | 15 | 18 | 46 | 27.96 | 15 | 23 | 37 | 107 | 38.43 | 14 | 23 | 54 | 187 |
| breakout | 11.86 | 7 | 11 | 15 | 43 | 10.64 | 7 | 9 | 13 | 90 | 9.49 | 7 | 9 | 11 | 83 |
| chopper command | 12.03 | 7 | 10 | 15 | 49 | 15.58 | 8 | 12 | 20 | 141 | 13.13 | 7 | 10 | 16 | 228 |
| crazy climber | 17.90 | 11 | 17 | 24 | 50 | 36.12 | 15 | 34 | 51 | 147 | 61.50 | 15 | 51 | 101 | 288 |
| demon attack | 9.52 | 6 | 8 | 11 | 49 | 11.17 | 7 | 10 | 13 | 135 | 9.88 | 6 | 8 | 12 | 222 |
| freeway | 15.03 | 10 | 14 | 19 | 48 | 27.56 | 15 | 21 | 35 | 135 | 44.50 | 17 | 34 | 60 | 210 |
| frostbite | 23.28 | 12 | 19 | 34 | 50 | 31.05 | 15 | 25 | 41 | 147 | 44.56 | 12 | 21 | 55 | 294 |
| gopher | 12.98 | 10 | 13 | 15 | 46 | 17.01 | 12 | 16 | 21 | 114 | 15.16 | 11 | 14 | 18 | 145 |
| hero | 22.30 | 10 | 19 | 33 | 50 | 17.06 | 6 | 10 | 21 | 147 | 14.80 | 5 | 7 | 13 | 276 |
| jamesbond | 20.28 | 10 | 17 | 29 | 50 | 25.91 | 14 | 25 | 35 | 138 | 28.77 | 13 | 24 | 41 | 246 |
| kangaroo | 17.89 | 11 | 16 | 23 | 49 | 39.29 | 25 | 39 | 52 | 150 | 46.69 | 24 | 40 | 62 | 223 |
| krull | 10.29 | 7 | 9 | 12 | 48 | 15.24 | 9 | 13 | 19 | 64 | 21.65 | 12 | 17 | 22 | 204 |
| kung fu master | 15.83 | 12 | 15 | 19 | 49 | 26.10 | 19 | 26 | 32 | 84 | 27.27 | 16 | 26 | 36 | 138 |
| ms pacman | 14.87 | 8 | 12 | 20 | 49 | 21.51 | 10 | 17 | 30 | 135 | 23.58 | 9 | 15 | 31 | 169 |
| pong | 23.21 | 15 | 23 | 31 | 50 | 50.28 | 28 | 48 | 67 | 144 | 67.51 | 19 | 60 | 102 | 264 |
| private eye | 10.61 | 4 | 6 | 16 | 50 | 8.22 | 5 | 7 | 10 | 129 | 6.22 | 2 | 5 | 8 | 168 |
| qbert | 13.09 | 7 | 11 | 17 | 48 | 26.83 | 15 | 25 | 36 | 138 | 38.31 | 13 | 30 | 59 | 228 |
| road runner | 8.17 | 4 | 5 | 10 | 49 | 10.14 | 5 | 7 | 10 | 120 | 6.61 | 4 | 5 | 6 | 139 |
| seaquest | 10.94 | 8 | 10 | 12 | 49 | 12.34 | 8 | 10 | 13 | 114 | 11.43 | 7 | 9 | 13 | 115 |
| up n down | 10.40 | 7 | 10 | 13 | 49 | 17.42 | 11 | 16 | 22 | 150 | 13.91 | 9 | 13 | 18 | 288 |
| Average | 14.52 | 8.77 | 12.58 | 18.85 | 48.54 | 20.74 | 11.65 | 18.23 | 26.69 | 121.46 | 24.92 | 10.58 | 19.35 | 33.58 | 197.58 |

Table 11: Prediction accuracy between options and environmental actions for $\ell_3$ and $\ell_6$ in 26 Atari games.

| Game | $\ell_3$ | | | $\ell_6$ | | | | | |
| --- | --- | --- | --- | --- | --- | --- | --- | --- | --- |
| | 0-step | 1-step | 2-step | 0-step | 1-step | 2-step | 3-step | 4-step | 5-step |
| alien | 73.43% | 81.24% | 66.76% | 68.84% | 75.00% | 61.55% | 56.89% | 53.50% | 51.49% |
| amidar | 75.55% | 81.32% | 71.78% | 78.02% | 82.90% | 62.28% | 55.19% | 50.08% | 48.32% |
| assault | 76.61% | 83.70% | 35.49% | 76.55% | 83.98% | 37.41% | 11.58% | 3.94% | 1.48% |
| asterix | 64.41% | 71.68% | 29.46% | 63.54% | 69.28% | 13.08% | 7.10% | 4.16% | 2.59% |
| bank heist | 91.84% | 93.82% | 72.82% | 90.59% | 91.42% | 55.11% | 35.85% | 23.09% | 15.64% |
| battle zone | 59.31% | 68.10% | 35.97% | 53.03% | 62.06% | 33.77% | 22.97% | 18.79% | 16.86% |
| boxing | 87.66% | 92.72% | 76.38% | 85.41% | 89.93% | 64.36% | 51.47% | 45.70% | 41.32% |
| breakout | 65.61% | 72.96% | 33.90% | 58.44% | 65.56% | 19.29% | 5.35% | 2.77% | 1.80% |
| chopper command | 71.11% | 76.93% | 51.90% | 69.16% | 74.82% | 33.97% | 23.98% | 19.02% | 13.41% |
| crazy climber | 84.24% | 91.40% | 77.09% | 83.95% | 89.90% | 67.90% | 57.60% | 53.24% | 50.56% |
| demon attack | 76.62% | 84.59% | 53.52% | 73.07% | 81.17% | 35.98% | 19.92% | 12.17% | 8.97% |
| freeway | 76.15% | 85.47% | 83.13% | 80.20% | 88.91% | 80.89% | 77.34% | 75.54% | 74.57% |
| frostbite | 73.53% | 80.95% | 75.86% | 68.05% | 75.05% | 67.44% | 62.74% | 59.82% | 57.79% |
| gopher | 85.18% | 90.50% | 71.23% | 80.08% | 86.74% | 41.73% | 28.23% | 20.99% | 17.57% |
| hero | 60.77% | 69.45% | 56.73% | 60.65% | 67.48% | 47.38% | 41.79% | 39.61% | 37.74% |
| jamesbond | 80.04% | 85.72% | 70.20% | 76.12% | 82.04% | 63.76% | 55.58% | 52.27% | 49.35% |
| kangaroo | 88.71% | 91.89% | 84.32% | 86.98% | 90.48% | 76.48% | 68.02% | 61.90% | 59.13% |
| krull | 73.75% | 81.45% | 49.56% | 75.78% | 82.41% | 57.58% | 35.83% | 24.81% | 19.37% |
| kung fu master | 85.54% | 90.36% | 77.14% | 84.72% | 89.95% | 58.19% | 47.24% | 39.70% | 36.20% |
| ms pacman | 71.76% | 78.05% | 68.81% | 70.02% | 79.43% | 49.92% | 40.89% | 37.39% | 36.39% |
| pong | 90.04% | 94.63% | 89.50% | 86.81% | 92.52% | 80.49% | 74.85% | 69.73% | 66.69% |
| private eye | 53.14% | 61.30% | 18.11% | 47.93% | 61.86% | 21.19% | 12.62% | 9.67% | 7.98% |
| qbert | 76.17% | 82.71% | 79.03% | 75.21% | 83.02% | 72.12% | 63.94% | 59.87% | 57.88% |
| road runner | 68.14% | 71.35% | 39.60% | 65.75% | 73.07% | 34.80% | 26.09% | 21.45% | 19.21% |
| seaquest | 76.62% | 82.36% | 37.48% | 74.19% | 80.98% | 35.47% | 21.45% | 15.91% | 12.74% |
| up n down | 80.72% | 88.11% | 77.24% | 77.96% | 87.10% | 56.07% | 43.43% | 34.44% | 30.06% |
| Average | 75.64% | 82.03% | 60.89% | 73.50% | 80.27% | 51.09% | 40.31% | 34.98% | 32.12% |

## D.4 PREDICTION ACCURACY OF OPTIONS

In this experiment, we check how the options suggested by the MCTS process predict future actions. For example, if an option *R-R-R* is suggested by the search at time $t$, we calculate the prediction accuracy using the recorded actions $a_t$, $a_{t+1}$, and $a_{t+2}$. The results for both $\ell_3$ and $\ell_6$ are shown in Table 11. As expected, the prediction accuracy decreases as the number of steps increases. Interestingly, the accuracy of the 1-step is higher than that of the 0-step. We assume this phenomenon is caused by softmax selection for options with repeated primitive actions. For example, the agent may apply *U* than the suggested options *R-R-R*. When this happens, at step $t + 1$, there is likely a much higher probability of acting *R* again to prompt the original decision. On the other hand, the prediction accuracy of $\ell_6$ is generally lower than those of $\ell_3$ at the same step number. It is hypothesized that increasing the maximum option length also increases the difficulty of training prediction and dynamics networks, thereby lowering the accuracy. To summarize, this observation verifies that the learned options closely correspond to the probabilities of choosing primitive actions.

### D.5 Top frequently used options

Finally, we investigate the distribution of frequently used options for each game. The accumulated usages are summarized in Table 12. Additionally, we extract the top 3 frequently used options for both $\ell_3$ and $\ell_6$ in Table 13 and Table 14. Generally speaking, the distribution of frequently used options is similar between $\ell_3$ and $\ell_6$, where the distribution is highly unbalanced, with the top 25% options accounting for over 95% of the usage. The most extreme case occurs in *freeway*, where the top 1% (options with repeated *U*) takes a proportion of more than 80%. On the other hand, for *crazy climber* and *ms pacman*, the top 1% usages have both been significantly increased from $\ell_3$ to $\ell_6$. However, the increase in usage does not correlate to the performance, as $\ell_6$ outperforms $\ell_3$ in *crazy climber* but underperforms in *ms pacman*, as shown in Table 1. Notably, we observe that from $\ell_3$ to $\ell_6$, the numbers of discovered options only increase by a factor of 2.43 on average, implying that there are not so many options practical for the planning. This analysis demonstrates that a small proportion of frequently used options play important roles in gameplay, and the number of discovered options will not grow excessively when relaxing the limitation of maximum option length.

Table 12: Accumulated usages of the frequently used options for $\ell_3$ and $\ell_6$ in 26 Atari games.

| Game | $\ell_3$ | | | | $\ell_6$ | | | |
|---|---|---|---|---|---|---|---|---|
| | Top 1% | Top 5% | Top 10% | Top 25% | Top 1% | Top 5% | Top 10% | Top 25% |
| alien | 21.67% | 74.01% | 89.35% | 98.64% | 53.56% | 81.47% | 90.76% | 98.15% |
| amidar | 41.82% | 91.33% | 98.26% | 99.50% | 49.67% | 83.14% | 94.95% | 99.23% |
| assault | 29.76% | 70.26% | 85.88% | 97.12% | 45.96% | 81.53% | 91.81% | 98.31% |
| asterix | 30.27% | 75.00% | 90.57% | 97.38% | 36.46% | 77.98% | 88.22% | 97.86% |
| bank heist | 22.46% | 50.28% | 71.80% | 96.60% | 20.68% | 54.44% | 70.90% | 90.26% |
| battle zone | 31.80% | 74.53% | 95.23% | 99.31% | 41.56% | 78.69% | 91.88% | 98.58% |
| boxing | 39.55% | 59.41% | 71.47% | 89.90% | 29.40% | 59.22% | 73.56% | 90.43% |
| breakout | 23.33% | 61.10% | 79.71% | 95.22% | 39.69% | 75.29% | 88.13% | 97.98% |
| chopper command | 40.51% | 90.41% | 96.61% | 99.33% | 35.91% | 75.72% | 91.05% | 99.05% |
| crazy climber | 18.00% | 52.19% | 73.53% | 95.66% | 52.62% | 82.73% | 91.65% | 98.26% |
| demon attack | 36.88% | 81.62% | 91.80% | 97.69% | 46.19% | 82.89% | 92.90% | 98.65% |
| freeway | 87.63% | 91.25% | 93.94% | 97.71% | 81.13% | 89.75% | 94.46% | 98.51% |
| frostbite | 58.73% | 90.74% | 96.43% | 99.21% | 62.58% | 86.18% | 93.24% | 98.26% |
| gopher | 58.17% | 77.73% | 86.00% | 96.81% | 46.93% | 83.74% | 92.58% | 98.51% |
| hero | 57.14% | 91.52% | 96.54% | 99.17% | 52.72% | 83.91% | 91.80% | 97.92% |
| jamesbond | 51.74% | 85.34% | 93.23% | 98.43% | 55.25% | 84.53% | 93.46% | 98.36% |
| kangaroo | 43.29% | 71.76% | 81.93% | 94.30% | 46.48% | 74.34% | 85.05% | 95.85% |
| krull | 23.50% | 56.37% | 74.32% | 93.80% | 21.42% | 54.80% | 73.48% | 92.43% |
| kung fu master | 48.52% | 79.10% | 89.64% | 97.27% | 46.32% | 75.97% | 86.49% | 96.42% |
| ms pacman | 28.04% | 88.11% | 96.09% | 99.22% | 41.82% | 75.41% | 90.17% | 98.02% |
| pong | 59.22% | 80.07% | 87.07% | 95.88% | 49.13% | 75.00% | 86.40% | 96.14% |
| private eye | 37.06% | 68.97% | 84.19% | 97.25% | 33.63% | 69.21% | 81.91% | 94.19% |
| qbert | 53.07% | 94.42% | 97.80% | 99.69% | 67.30% | 91.71% | 96.82% | 99.30% |
| road runner | 28.77% | 80.77% | 90.97% | 98.11% | 42.47% | 78.51% | 89.48% | 97.43% |
| seaquest | 29.84% | 67.32% | 80.34% | 94.60% | 36.79% | 73.06% | 86.27% | 96.69% |
| up n down | 62.27% | 85.91% | 94.45% | 98.97% | 53.55% | 86.24% | 94.25% | 99.14% |
| Average | 40.89% | 76.52% | 87.97% | 97.18% | 45.74% | 77.52% | 88.53% | 97.07% |

Table 13: The top 3 frequently used options with their total share for $\ell_3$ in 26 Atari games.

| Game | Top 1 | Top 2 | Top 3 | % |
|---|---|---|---|---|
| alien | UR-UR-UR | LF-LF-LF | UL-UL-UL | 10.01% |
| amidar | LF-LF-LF | R-R-R | DF-DF-DF | 18.07% |
| assault | R-R | L-L | R-R-R | 9.07% |
| asterix | U-U-U | D-D | UL-UL | 2.79% |
| bank heist | L-R-L | L-UR-L | DL-UR-DL | 16.79% |
| battle zone | DF-DF-DF | DRF-DRF-DRF | LF-LF-LF | 3.53% |
| boxing | R-R-R | DR-DR-DR | DRF-DRF-DRF | 23.92% |
| breakout | N-N-N | N-N | L-L-L | 10.52% |
| chopper command | DF-DF-DF | UF-UF-UF | ULF-ULF-ULF | 9.69% |
| crazy climber | D-D-U | U-U-U | U-U-D | 12.16% |
| demon attack | R-R-R | L-L-L | LF-LF-LF | 10.77% |
| freeway | U-U-U | N-N-N | U-U | 57.12% |
| frostbite | D-D-D | UR-UR-UR | DR-DR-DR | 24.61% |
| gopher | L-L-L | R-R-R | RF-RF-RF | 26.55% |
| hero | R-R-R | DLF-DLF-DLF | LF-LF-LF | 8.11% |
| jamesbond | UR-UR-UR | L-DLF-L | L-L-L | 21.31% |
| kangaroo | D-D-D | F-F-F | LF-LF-LF | 30.37% |
| krull | DL-DL-DL | UR-UR-UR | DL-DL | 10.17% |
| kung fu master | DR-DR-DR | DLF-DLF-DLF | ULF-ULF-ULF | 20.89% |
| ms pacman | UL-UL-UL | DL-DL-DL | UR-UR-UR | 14.52% |
| pong | LF-LF-LF | F-F-F | N-N-N | 48.94% |
| private eye | URF-URF | L-L-L | L-L | 2.37% |
| qbert | D-D-D | R-R-R | L-L-L | 36.60% |
| road runner | DLF-DLF-DLF | UL-UL-UL | DL-DL-DL | 5.82% |
| seaquest | DF-DF-DF | DLF-DLF-DLF | DLF-DLF | 6.12% |
| up n down | U-U-U | UF-UF-UF | DF-DF-DF | 34.05% |

Table 14: The top 3 frequently used options with their total share for $\ell_6$ in 26 Atari games.

| Game | Top 1 | Top 2 | Top 3 | % |
|---|---|---|---|---|
| alien | DR-DR-DR-DR-DR-DR | URF-URF-URF-URF-URF-URF | LF-LF-LF-LF-LF-LF | 13.52% |
| amidar | LF-LF-LF-LF-LF-LF | UF-UF-UF-UF-UF-UF | RF-RF-RF-RF-RF-RF | 13.30% |
| assault | R-R | L-L | U-L-L | 8.35% |
| asterix | UL-UL | UR-UR | D-D | 3.49% |
| bank heist | DR-L | UR-L | R-DL | 6.77% |
| battle zone | DF-DF-DF-DF-DF-DF | UL-UL-UL-UL-UL-UL | U-U | 2.43% |
| boxing | DR-DR-DR-DR-DR-DR | DR-DR | N-N-N-N-N-N | 9.83% |
| breakout | F-F | N-N | L-L | 9.75% |
| chopper command | UF-UF | DF-DF | DLF-DLF-DLF-DLF-DLF-DLF | 5.29% |
| crazy climber | U-U-DL-DL-DL-U | UL-UL-DR-DL-DL-UL | U-U-DL-DL-DR-U | 14.20% |
| demon attack | RF-RF | L-L | LF-LF | 6.60% |
| freeway | U-U-U-U-U-U | U-U | D-U-U-U-U-U | 45.91% |
| frostbite | F-F-F-F-F-F | DF-DF-DF-DF-DF-DF | DRF-DRF-DRF-DRF-DRF-DRF | 14.63% |
| gopher | L-L | R-R | R-R-R-R-R-R | 11.25% |
| hero | DLF-DLF-DLF-DLF-DLF-DLF | RF-RF-RF-RF-RF-RF | DRF-DRF-DRF-DRF-DRF-DRF | 3.65% |
| jamesbond | DL-DL-DL-DL-DL-DL | LF-LF-LF-LF-LF-LF | UR-UR-UR-UR-UR-UR | 11.66% |
| kangaroo | R-R-R-R-R-R | L-L-L-L-L-L | DL-DL-DL-DL-DL-DL | 17.11% |
| krull | UR-UR-UR-UR-UR-UR | DL-DL | L-L-L-L-L-L | 4.40% |
| kung fu master | DR-DR-DLF-DR-DR-DR | L-L-L-L-L-L | URF-URF-URF-URF-URF-URF | 7.57% |
| ms pacman | UL-UL-UL-UL-UL-UL | UL-UL | DR-DR-DR-DR-DR-DR | 9.56% |
| pong | RF-RF-RF-RF-RF-RF | R-R-R-R-R-R | LF-LF-LF-LF-LF-LF | 24.58% |
| private eye | URF-URF | ULF-ULF | DF-DF-DF-DF-DF-DF | 0.78% |
| qbert | F-F-F-F-F-F | R-R-R-R-R-R | U-U-U-U-U-U | 21.94% |
| road runner | ULF-ULF-ULF-ULF-ULF-ULF | L-L | L-L-L-L-L-L | 1.62% |
| seaquest | DLF-DLF | DRF-DRF | DLF-DLF-DLF-DLF-DLF-DLF | 3.30% |
| up n down | U-U-U-U-U-U | DF-DF | U-U | 15.18% |

