# OpenReview forum: "OptionZero: Planning with Learned Options"
_ICLR.cc/2025/Conference — ICLR 2025 Oral_

### Official Review · Reviewer_jR7c · 2024-10-30

**Soundness:** 4
**Presentation:** 4
**Contribution:** 4
**Rating:** 8
**Confidence:** 4

**Summary:**

This paper introduces the OptionZero framework, which incorporates options into MCTS and enables the automatic design of options through self-play, thereby avoiding cumbersome manual design. In Atari games, OptionZero achieves significant improvements compared to MuZero, achieving a 131.58% improvement in mean human-normalized score. It has shown promising directions for discovering and using options in planning.

**Strengths:**

1. This paper is well-written with a clear structure, making the research content easily comprehensible.

2. By introducing the OptionZero framework and leveraging self-play for automatic option design, this study paves the way for new approaches. The novelty is good.

3. The experimental results robustly support the effectiveness of the algorithm.

**Weaknesses:**

1. I am curious about whether the introduction of options has any impact on the theoretical optimality of MCTS.

2. Is it possible to demonstrate the advancement of the algorithm more directly by solely utilizing the learned network for action selection, without relying on MCTS?

3. What is the expected performance as the value of $l$ increases?

4. What are the differences in the running wall-clock time required for OptionZero compared to MuZero?

**Questions:**

Please refer to the weakness part.

---

> ### Author Response · Authors · 2024-11-21
> **Respond to Reviewer jR7c**
>
> We thank the reviewer for providing thoughtful and insightful feedback. We answer each question below.
>
> > I am curious about whether the introduction of options has any impact on the theoretical optimality of MCTS.
>
> We carefully designed OptionZero to preserve the theoretical optimality of MCTS. To achieve this, we maintain the statistical consistency ($N$, $P$, $Q$, and $R$ in edges) and add an additional link for option edges without altering MCTS's theoretical guarantees. Specifically, the primitive selection is designed to align with the original search direction, followed by option selection. This approach allows OptionZero to maintain the exploration-exploitation balance by the PUCT formula, ensuring that the introduction of options maintains the asymptotic convergence properties of MCTS.
>
> > Is it possible to demonstrate the advancement of the algorithm more directly by solely utilizing the learned network for action selection, without relying on MCTS?
>
> Indeed, our option network, designed to learn the dominant options, can be utilized independently of MCTS. For example, it could be integrated into other reinforcement learning algorithms, such as PPO or SAC, which do not rely on search-based planning. However, our primary motivation was not only to enable the autonomous discovery of options adapted to different states but also to address the computational challenges of using primitive actions in search, which require extensive simulations for deeper searches. This is why we focus on integrating options with MuZero planning.
>
> We appreciate the reviewer for highlighting this intriguing direction, which offers valuable potential for future exploration. Thank you for your insightful comment!
>
> > What is the expected performance as the value of $l$ increases?
>
> Our empirical results indicate that both $\ell_3$ and $\ell_6$ outperform the baseline $\ell_1$, with $\ell_3$ slightly outperforming $\ell_6$ in Atari games. This difference may arise from two factors: (1) increasing the maximum option length adds complexity to the space of possible options, requiring longer adaptation for both the option and dynamics networks, and (2) with a frameskip of 4, an option of length 6 effectively skips 24 frames, which also further increases the learning difficulty.
>
> In our experiments, we find $\ell_3$ or $\ell_6$ sufficient for the Atari environment, with performance likely converging or slightly declining as $l$ increases. However, this does not restrict OptionZero to option lengths of 3 or 6 in all environments. For instance, in our toy environment GridWorld, an option length of 9 successfully discovers the optimal path, suggesting that longer option lengths could be effective in other environments.
>
> > What are the differences in the running wall-clock time required for OptionZero compared to MuZero?
>
> Please refer to the general response for all reviewers.

---

### Official Review · Reviewer_S6xr · 2024-10-31

**Soundness:** 3
**Presentation:** 3
**Contribution:** 2
**Rating:** 6
**Confidence:** 4

**Summary:**

The authors introduce OptionZero, an advanced extension of the MuZero algorithm that autonomously identifies temporally extended actions, known as options, without the need for predefined options or expert-provided demonstrations. By incorporating an option network, OptionZero enhances planning efficiency by decreasing both decision-making frequency and the computational load required for complex environments. Evaluated on a suite of Atari games, OptionZero demonstrates notable improvements in human-normalized scores compared to MuZero, accompanied by a detailed analysis of how options are utilized across varying game states and scenarios

**Strengths:**

- Novel idea for autonomous and adaptable option discovery. The proposed method's ability to autonomously discover and tailor options to diverse game dynamics removes the need for predefined actions, making it highly adaptable across different environments.

- Convincing results for enhanced planning in RL.  By integrating an option network, OptionZero reduces decision frequency, enabling computational efficiency, particularly in visually complex tasks like Atari games.

- Strong Performance Gains: On Atari benchmarks, OptionZero achieves a 131.58% improvement over MuZero, a significant improvement compared to previous SOTA papers.

- Interesting ideas to adjust option lengths, balancing performance and training efficiency, particularly useful in tasks needing variable action sequences.

**Weaknesses:**

- Inconsistent Option Use Across Games: OptionZero's reliance on options appears to vary widely across Atari games. While longer options bring substantial gains in some games, they contribute less in others. This inconsistency suggests that the model’s option-based planning may struggle to generalize well across diverse, complex environments. The paper should discuss this limitation.

- Challenges in Complex Action Spaces: In games with intricate action spaces, such as Bank Heist (Atari), OptionZero’s dynamic network encounters difficulty as option lengths increase, particularly with multi-step dependencies. This issue may restrict OptionZero’s application in environments where actions are highly combinatorial, relying instead on settings with more straightforward or predictable actions.

- Reduced Prediction Accuracy for Longer Options: The model’s prediction accuracy tends to decrease as options become longer, affecting planning quality where extended strategies are essential. I would recommend adding an experiment to study this effect and discuss potential limitations of the proposed method.


- Limited Application Beyond Games: Although the model shows promise in game environments, the paper does not investigate its potential beyond Atari-like settings. I would appreciate seeing results on other domains, maybe robotic such as Gymnasium-Robotics. Under the current evaluation, the method seems limited to game-based scenarios.

**Questions:**

- How does the dynamics network handle complex action spaces, especially in games with highly varied option paths?"

- What are the specific computational costs of incorporating the option network, both in training and during MCTS simulations? Could the author discuss the associated overhead with the proposed method?

- Can the model be applied to environments beyond games with less predictable state transitions, and how would option discovery be affected? I would suggest adding studies in robotic environments.

---

> ### Author Response · Authors · 2024-11-21
> **Respond to Reviewer S6xr**
>
> We thank the reviewer for providing thoughtful and insightful feedback. We answer each question below.
>
> > Inconsistent Option Use Across Games
>
> Overall, using options improves performance, as the human-normalized mean for both $\ell_3$ and $\ell_6$ are higher than for $\ell_1$. However, longer options may not always contribute to better performance in every environment. We have conducted several analyses to investigate the underlying reasons, as detailed in Appendix D. However, our findings suggest that the performance is influenced by a combination of factors, with no clear correlations observed, leaving the exact reasons open for further investigation. The revised version addresses these challenges in the discussion section.
>
> > Challenges in Complex Action Spaces
>
> Appendix D.1 shows the number of option types discovered in each game. One might expect that a high number of option types could increase the complexity of OptionZero's learning and reduce performance. For example, in jamesbond, it contains 376 and 735 option types in $\ell_3$ and $\ell_6$, and $\ell_3$ performs better than $\ell_6$. However, in kung fu master, although it contains numerous option types (536 and 1386), $\ell_6$ still slightly outperforms $\ell_3$, implying other positive factors contribute to learning. Similar to our response to the previous question, the performance is influenced by a combination of factors.
>
> > Reduced Prediction Accuracy for Longer Options
>
> > How does the dynamics network handle complex action spaces, especially in games with highly varied option paths?
>
> Complex action spaces indeed increase the learning complexity for the dynamics network in MuZero, a challenge highlighted in several papers. Our work mainly focuses on investigating the method for autonomous discovering options and utilizing them during planning. Hence, we do not introduce specific adaptations for the complex action spaces but train them using the same approach as in MuZero. To address this challenge, the dynamics network could be further improved by integrating with other techniques for future works, such as S4 [1] or Dreamer [2]. However, this is beyond the scope of this paper. We have addressed this as a future direction in the discussion.
>
> [1] Gu, Albert, Karan Goel, and Christopher Re. "Efficiently Modeling Long Sequences with Structured State Spaces." International Conference on Learning Representations.
>
> [2] Hafner, Danijar, et al. "Dream to Control: Learning Behaviors by Latent Imagination." International Conference on Learning Representations.
>
> > Limited Application Beyond Games
>
> > Can the model be applied to environments beyond games with less predictable state transitions, and how would option discovery be affected? I would suggest adding studies in robotic environments.
>
> MuZero is a powerful zero-knowledge learning algorithm that achieves high performance in games. For environments with less predictable state transitions, such as robotic environments, Sampled MuZero [3], an extension of MuZero, has been developed to handle complex action spaces effectively. It would be a promising future direction to extend OptionZero to such complex environments by integrating it with Sampled MuZero. However, as OptionZero is built upon MuZero, which is designed specifically for games, incorporating Sampled MuZero is nontrivial to accomplish within the rebuttal period. We leave this as a direction for future work.
>
> [3] Hubert, Thomas, et al. "Learning and planning in complex action spaces." International Conference on Machine Learning. PMLR, 2021.
>
> > What are the specific computational costs of incorporating the option network, both in training and during MCTS simulations?
>
> Please refer to the general response for all reviewers.

---

### Official Review · Reviewer_EWix · 2024-11-03

**Soundness:** 3
**Presentation:** 3
**Contribution:** 3
**Rating:** 8
**Confidence:** 3

**Summary:**

This paper presents OptionZero, a novel approach that incorporates an option network into MuZero and allows the agent to learn temporary extended actions. The authors conducted empirical experiments and find OptionZero outperforms MuZero in teams of mean human-normalized scores on 26 Atari games.

**Strengths:**

The paper is well-written. The authors explain the use of options clearly with a toy example, demonstrating how options are used. The empirical results are also strong, achieving high mean normalized scores.

**Weaknesses:**

It's not clear the actual benefits options bring. In the intro, the paper claims options allow for "searching deeper", but the empirical analysis shows "deeper search is likely but not necessary for improving performance". While it's nice to have the option to do option, could the authors provide a more detailed analysis of options beyond a deeper search?

The paper could also benefit more from discussions of
1) the trade-offs between increased complexity and performance gains
2) how much tuning did the authors perform to make OptionZero work; were there failure cases/ideas during the development and how did the authors overcome them?

**Questions:**

1. Why select these 26 Atari games instead of using the standard 57 Atari games?
1. What is the hardware resource for conducting the experiments?
1. How long does training a single Atari game with OptionZero take?
1. How long does training a single Atari game with MuZero take?

---

> ### Author Response · Authors · 2024-11-21
> **Respond to Reviewer EWix**
>
> We thank the reviewer for providing thoughtful and insightful feedback. We answer each question below.
>
> > It's not clear the actual benefits options bring. In the intro, the paper claims options allow for "searching deeper", but the empirical analysis shows "deeper search is likely but not necessary for improving performance".
>
> This is a good point. To further investigate the behavior of deep search, besides the average and maximum tree depths, we also examine the median tree depth. Interestingly, we observe that in certain games, such as asterix, battle zone, hero, and private eye, deep searches occur only in specific states. This suggests that the search depth varies depending on the requirements of different states. We have revised Section 5.4 to include these findings and added the median results to Table 10 and Appendix D.3. Thank you again for highlighting this!
>
> > While it's nice to have the option to do option, could the authors provide a more detailed analysis of options beyond a deeper search?
>
> Certainly, we understand that it is interesting to explore the behavior of options. In fact, we have included several analyses in Appendix D, including the proportions of options applied in each game (D.1), how longer options are discovered during the training process (D.2), the detailed proportions of options used within the search (D.3), how the options suggested by the MCTS process predict future actions (D.4), and the distribution of frequently used options (D.5). These analyses provide deeper insights into the learned models from various perspectives, illustrating how options are utilized in different games. Due to the page limitations, we present only a subset of these results in the main text, with the full details available in the appendix.
>
> > the trade-offs between increased complexity and performance gains
>
> > How long does training a single Atari game with OptionZero take?
>
> > How long does training a single Atari game with MuZero take?
>
> Please refer to the general response for all reviewers.
>
> > were there failure cases/ideas during the development and how did the authors overcome them?
>
> Yes, one of the most challenging aspects was integrating options into MCTS. In our early trials, we considered expanding an option node directly as a single child node instead of creating all internal nodes and using an option edge to link across multiple nodes. However, this led to inconsistencies in the statistical information – for instance, the action sequence "UP-UP-UP" could result from either executing three consecutive primitive "UP" nodes or a single "UP-UP-UP" node, making it difficult for MCTS to maintain statistical consistency. This is the reason why, in our design, we preserve all nodes in the original search tree, create option edges, and split selection into two stages to ensure that statistical information remains accurate.
>
> Since these failure experiments are incomplete and unsuitable even as baselines, and due to page limitations, we only present the final successful method in our paper.
>
> > Why select these 26 Atari games instead of using the standard 57 Atari games?
>
> The primary reason for using 26 games instead of 57 games is the constraint on computational resources. Training a single model on one Atari game requires approximately 22 hours. Reproducing the results in Table 1 requires about 214.5 days of training time on a single machine (3 models * 26 games * 3 seeds per game * 22 hours). Therefore, we select 26 Atari games, drawn from SimPLe [1], which are widely recognized as benchmarks for evaluating performance in Atari games.
>
> [1] Kaiser, Lukasz, et al. "Model Based Reinforcement Learning for Atari." International Conference on Learning Representations.
>
> > What is the hardware resource for conducting the experiments?
>
> The hardware resources used for our experiments are detailed in Appendix B. Specifically, all experiments are conducted on machines with 24 CPU cores and four NVIDIA GTX 1080 Ti GPUs.

---

> > ### Comment · Reviewer_EWix · 2024-11-24
> > **Reply**
> >
> > Thanks for these clarifications. I have updated the scores correspondingly.

---

> > > ### Author Response · Authors · 2024-11-26
> > >
> > > Thank you for raising the scores! We are happy that our clarifications address your questions.

---

### Official Review · Reviewer_JCLU · 2024-11-04

**Soundness:** 3
**Presentation:** 3
**Contribution:** 4
**Rating:** 8
**Confidence:** 3

**Summary:**

The work proposes a revamped approach to the well-known options framework, which allows agents to take temporally extended actions as well as myopic ones. By combining a network which learns options with MCTS MuZero (which models transition dynamics) the authors propose a method to utilise options alongside single actions in self-play games.

**Strengths:**

The need for efficiency in decision making in RL is clear, as single-step actions are slow and computationally expensive (even more so in slow simulators). Thus, the problem addressed by OptionZero is clear and its existence is well-motivated. Additionally, since much prior work in options appear to be in manually defined and demonstration-based settings, the generalisability of OptionZero is a strong selling point.

Within fixed computational constraints, the idea of decreasing the frequency of queries to the network is a strong idea for the current state of RL. Also important is the notion of learning subroutines which the options network will identify as useful in different scenarios and not have to re-learn temporal relationships.

The flexibility to play options or primitive actions results in tailored reactions to scenarios, as an agent may need the fine-grained approach taken by traditional RL. The main results in Table 1 indicate the validity of the method, as using options provides a performance benefit more often than not, with longer option limits sometimes outperforming shorter ones.

**Weaknesses:**

It is unclear why options are outperformed by primitive actions in certain environments. The authors suggest that in environments with high combinatorial complexity, learning of the dynamics model may be difficult and thus options may simply produce more overhead than actual benefit. A more detailed analysis of these environments would be beneficial, for e.g. investigate whether there is a correlation between the stochastic branching factor of the environment and the performance of options.

Additionally, it seems that longer options may improve efficiency but not always increase performance when those options may be overextending in environments where more granular control is required. Have the authors considered implementing dynamic options lengths somehow? This may make the idea more viable, or at least a discussion on the complications of implementing that would be a good addition to the work.

**Questions:**

Clerical: In section 5.2 the $l_{1}$ option setting is mentioned as a baseline, but Table 1 compared the options to something called $l_{0}$. Do $l_{0}$ and $l_{1}$ refer to the same baseline? If so, using consistent notation will help make the results section more readable.

---

> ### Author Response · Authors · 2024-11-21
> **Respond to Reviewer JCLU**
>
> We thank the reviewer for providing thoughtful and insightful feedback. We answer each question below.
>
> > It is unclear why options are outperformed by primitive actions in certain environments … A more detailed analysis of these environments would be beneficial
>
> > longer options may improve efficiency but not always increase performance
>
> We appreciate the reviewer's insightful observation. In fact, we share the curiosity about why options outperformed primitive actions in certain environments and have conducted several analyses to investigate the underlying reasons, as detailed in Appendix D.
>
> Our experiments reveal no single factor that directly correlates with performance. Instead, the performance of a single game likely results from a combination of factors, which vary across games. For example, regarding the stochastic branching factor, Appendix D.1 shows the number of option types discovered in each game. In jamesbond, it contains 376 and 735 option types in $\ell_3$ and $\ell_6$, and $\ell_3$ performs better than $\ell_6$. This is likely due to increased action space complexity. However, in kung fu master, although it contains numerous option types (536 and 1386), $\ell_6$ still slightly outperforms $\ell_3$, implying other positive factors contribute to learning.
>
> We also explored other aspects, including how longer options are discovered during the training process (D.2), the detailed proportions of options used within the search (D.3), how the options suggested by the MCTS process predict future actions (D.4), and the distribution of frequently used options (D.5). While these analyses provide deeper insights, no clear correlations are observed, leaving the exact reasons open for further investigation.
>
> > Have the authors considered implementing dynamic options lengths somehow?
>
> We would like to clarify that our design already supports dynamic option lengths, allowing option lengths to vary from 1 to $L$, where $L$ is the predefined maximum length. Table 2 illustrates the distribution of different option lengths used under $L=3$ and $L=6$.
>
> As this is the first work to explore learning options and utilize learned options in MCTS, we choose a fixed maximum length $L$ to effectively demonstrate the concept. A potential future direction could involve starting with $L=3$ and then increasing $L$ to 6, or designing a scheduling mechanism to dynamically adjust the maximum option length. We have included this as a potential direction for future work in the discussion.
>
> > Do $l_0$ and $l_1$ refer to the same baseline?
>
> Yes, it should be $\ell_1$. We have corrected this in the revised version. Thank you for pointing it out!

---

> > ### Comment · Reviewer_JCLU · 2024-11-26
> >
> > Thank you for your response and your new discussion sections in the paper. I will leave the current review score as is, as I believe it is still appropriate.

---

> > > ### Author Response · Authors · 2024-12-03
> > >
> > > Thank you for your support and recommendation of our paper. We truly appreciate it!

---

### Author Response · Authors · 2024-11-21
**General Response and Summary of Revision**

Dear all reviewers,

We sincerely appreciate the reviewers' thoughtful comments and constructive feedback.

---

We would like to answer a general question raised by all reviewers regarding the complexity of OptionZero, the computational costs, and running wall-clock time for training OptionZero:

The increased complexity of OptionZero is negligible, as our approach does not require additional simulations or examination of internal states to expand an option node. The only added complexity involves (a) minimal computational costs for the option network, which is simply an additional head for predicting the dominant option, and (b) maintaining statistics for the option edges.

These costs are negligible compared to the overall MCTS process, as also confirmed by our experiments. Specifically, for $\ell_1$ (MuZero), the training time is approximately 21.89 hours. For $\ell_3$ and $\ell_6$ (OptionZero), the training times increase slightly to around 22.28 hours and 22.95 hours, representing increases of 1.8% and 4.8%, respectively.

We have revised subsection 4.2 and Appendix B to include these descriptions.

---

In addition, we have uploaded a revised version incorporating the suggested improvements. All revisions are highlighted in blue text for better clarity and ease of review. We summarize the changes as follows.

* Add algorithm complexity, computational costs, and specific training time in Section 4.2, Appendix A.1, and Appendix B. (Reviewer EWix, S6xr, jR7c)
* Add a discussion on median tree depth in Section 5.4, and provide the 25th, 50th, and 75th percentiles of the tree depth in Table 10 in Appendix D.3. (Reviewer EWix)
* Add discussions on limitations and future works in Section 6. (Reviewer JCLU, S6xr)
* Fix a typo in Table 1. (Reviewer JCLU)
* Reduce the size of Figure 2.

Thank you for your time and effort in reviewing our paper. Please let us know if you have any further suggestions.

---

### Meta-Review · Area_Chair_QW8Y · 2024-12-09

**Metareview:**

This paper integrates hierarchical planning into MCTS via the option framework. The algorithm is neat and the empirical performance improvement is quite significant. The analysis of the learned options is also very illustrating. One possible weakness is the lack of theoretical analysis of MCTS with options. Making options really work at scale has been in the wish list of the RL community for quite a long time, and I am glad to see that this paper finally delivers it. I, therefore, recommend oral presentation.

**Additional Comments On Reviewer Discussion:**

All reviewers unanimously agree this is a great paper.

---

### Decision · Program_Chairs · 2025-01-22

Accept (Oral)